# MUniverse: A Simulation and Benchmarking Suite for Motor Unit Decomposition

**Pranav Mamidanna**[1,2,*], **Thomas Klotz**[3,*], **Dimitrios Halatsis**[2,*],
**Agnese Grison**[2], **Irene Mendez-Guerra**[1,2], **Shihan Ma**[2], **Arnault H. Caillet**[2],
**Simon Avrillon**[4], **Robin Rohlén**[2,5], **Dario Farina**[2].

[1] I-X Center for AI in Science, Imperial College London, UK
[2] Department of Bioengineering, Imperial College London, UK
[3] Institute for Modelling and Simulation of Biomechanical Systems, University of Stuttgart, Germany
[4] Université Côte d'Azur, LAMHESS, Nice, France
[5] Department of Diagnostics and Intervention, Umeå University, Sweden
[*] These author have equally contributed to the work

## Abstract

Neural source separation enables the extraction of individual spike trains from complex electrophysiological recordings. When applied to electromyographic (EMG) signals, it provides a unique window into the motor output of the nervous system by isolating the spiking activity of motor units (MUs). MU decomposition from EMG signals is currently the only scalable neural interfacing approach available in behaving humans and has become foundational in motor neuroscience and neuroprosthetics. However, unlike related domains such as spike sorting or electroencephalography (EEG) analysis, the decomposition of EMG signals lacks open benchmarks that reflect the diversity of muscles, movement contexts, and noise sources encountered in practice. To address this gap, we introduce MUniverse, a modular simulation and benchmarking suite for decomposing EMG signals into individual MU spiking activity. MUniverse provides: (1) a simulation stack with a user-friendly interface to a state-of-the-art EMG generator; (2) a curated library of datasets across synthetic, hybrid synthetic-real data with ground truth spikes, and experimental EMG; (3) a set of internal and external decomposition pipelines; and (4) a unified benchmark with well-defined tasks, standard evaluation metrics, and baseline results from established decomposition pipelines. MUniverse is designed for extensibility, reproducibility, and community use, and all datasets are distributed with standardised metadata (Croissant, BIDS). By standardising evaluation and enabling dataset simulation at scale, MUniverse aims to catalyze progress on this long-standing neural signal processing problem.

## 1 Introduction

Modern neuroscience increasingly relies on source separation techniques that recover latent neural signals from mixtures, aligning with core machine learning challenges of inverse problems and representation learning. These neural (blind) source separation problems are ubiquitous across neuroscience, from spike sorting in extracellular electrophysiological recordings to calcium imaging demixing, and source localization in electroencephalography (EEG) and magnetoencephalography recordings. Each instance requires sophisticated algorithms to invert ill-posed signal models and uncover the underlying neural activity patterns. Electromyographic (EMG) recordings present a particularly compelling case for machine learning researchers: each surface electrode captures a convolutive mixture of motor unit (MU) action potentials propagating through biological tissues, creating a natural testbed for advanced blind source separation (BSS) techniques.

39th Conference on Neural Information Processing Systems (NeurIPS 2025) Track on Datasets and Benchmarks.

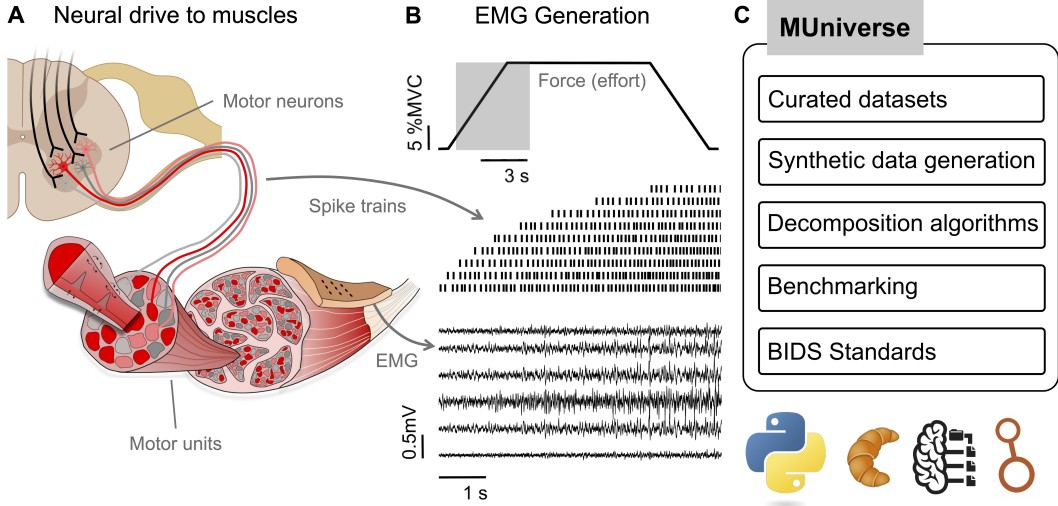

Figure 1: **MUniverse overview and motivation.** (**A**) Decomposition of EMG signals obtained at the surface of the skin into individual motor unit spike trains provides a unique window into the neural control of movement. (**B**) The generation pathway from spike trains to EMG during an isometric contraction with a ramp and hold pattern of effort exertion. Neural commands from the spinal cord (shown as a raster plot) drive the motor units to produce a particular pattern of force, and the resulting biopotentials are captured using HDsEMG. (**C**) The scope of MUniverse includes curated datasets, a simulation stack, algorithms, and tools to standardize EMG datasets.

The MU is the smallest voluntarily contractible unit comprising a motor neuron and an axon that innervates tens to hundreds of muscle fibres (Figure 1A). EMG-based decomposition into MU activity has played an important role in, e.g., improving our understanding of the neural control of movement [23], developing neural interfaces with applications such as prosthetic control [15] and human augmentation [12]. Decomposition – extracting individual MU activities from the mixed EMG signals – has therefore emerged as a foundational technique with broad scientific and clinical impact.

Over the past two decades, numerous BSS methods have been proposed to decompose high-density surface EMG (HDsEMG) signals into individual MU spike trains [22, 8, 37, 9, 19]. The gold standard for validating BSS methods is the two-source test, i.e., concurrent recording of HDsEMG and intramuscular needle or fine wire electrodes, comparing the rate of agreement between the two methods [14]. However, intramuscular fine wire electrodes provide a limited number of detectable MUs due to their spatial selectivity. Further, there are very few open datasets that include these simultaneous recordings. While insights into the performance of decomposition algorithms can be obtained from simulated data with known ground truth, such data may oversimplify certain aspects of real-world EMG signals. Despite a growing ecosystem of algorithms, EMG decomposition still lacks the rigorous benchmarking culture that catalysed breakthroughs in spike sorting [34] [5]. Methods are evaluated on private data, with heterogeneous pre-processing pipelines and different metrics, which makes it challenging to gauge real progress or identify failure modes.

The field of neural interfacing and motor neuroscience, therefore, needs a standardised, open, and extensible benchmark that spans simulated, hybrid, and experimental recordings.

**Our contributions**  To address this issue, we introduce MUniverse (Figure 1), an open-source simulation and benchmarking suite for decomposing EMG recordings into MU spike trains, including:

- A containerized simulation stack that integrates musculoskeletal biomechanics, motor neuron pool dynamics, and generative models of MU action potentials (MUAPs) for end-to-end generation of EMG signals.

- A library of curated, diverse datasets: fully simulated, hybrid (experimentally recorded MUAPs convolved with synthetic spike trains), and experimental EMG.

- A uniform API to a suite of decomposition algorithms, both implemented natively, as well as containerized wrappers to existing algorithms.
- And a standard evaluation framework for decomposition outputs, that generates a report card using uniform metrics of source quality and signal reconstruction.

By unifying data generation, evaluation, and reporting under FAIR principles, MUniverse enables the community to measure progress rigorously and accelerate the next decade of EMG-based neural interfacing. MUniverse is currently hosted on GitHub[1].

## 2 Background and related work

### 2.1 EMG signal model

EMG signals can be modelled as a linear convolutive mixture [13]:

$$\mathbf{x}(t) \; = \; \sum_{l=0}^{L-1} \mathbf{H}(l,t)\, \mathbf{s}(t-l) \; + \; \boldsymbol{\varepsilon}(t) \,, \tag{1}$$

where $t$ is a discrete time sample, $\mathbf{x}(t) \in \mathbb{R}^M$ is the EMG signal with $M$ being the number of channels, $\mathbf{H}(l,t) \in \mathbb{R}^{M \times N}$ contains the finite MU impulse responses (i.e., the MUAPs) of length $L$ samples, $\boldsymbol{\varepsilon}(t)$ is additive noise, and $\mathbf{s}(t) \in \{0,1\}^N$ are the $N$ binary MU spike trains. The spike train for MU $j$ is defined as $s_j(t) = \sum_{r \in \mathcal{S}_j} \delta(t - t_j^r)$ where $\mathcal{S}_j = \{t_j^1, ..., t_j^{T_j}\}$ is the set of discharge times and $\delta(\cdot)$ the Dirac delta function. Often $\mathbf{H}(l,t)$ is assumed to be stationary (e.g., by considering non-fatiguing and isometric contractions), i.e., not being dependent on time.

### 2.2 Decomposition methods

The goal of EMG decomposition is to estimate the MU spike trains $\mathbf{s}(t)$ only given the observed EMG signals $\mathbf{x}(t)$ (see Equation (1)). Existing decomposition methods can be classified into three groups: (1) template-matching, (2) convolutive BSS, and (3) deep learning-based methods.

**Template matching** summarizes a group of (semi-)supervised data analysis methods, which are well-established in many fields of electrophysiological spike sorting problems [e.g., 34]. In short, these methods find, match, and update MUAP templates (waveforms) [10], and resolve complex superpositions [36]. Well-known algorithms in this category are the PD III [10] and PD-IPUS + PD-IGAT [36], where the latter is an extension of the former. Yet, compared to invasive recordings, surface EMG signals have a smaller bandwidth due to low-pass filtering of the volume conductor, making the MUAPs more similar and the inverse problem more challenging. Hence, template matching methods often face limitations in the presence of multiple overlapping spikes (e.g., high-intensity contractions or doublet discharges [38]) or signal non-stationarities. Moreover, no open-source code exists for solving the HDsEMG-based decomposition problem using template-matching methods, leaving the scientific community uncertain regarding the validity of the obtained MU spike train estimates.

**Convolutive BSS** (CBSS) is currently the most common method for estimating MU spike trains from surface EMG signals. In CBSS, MU spike trains are estimated by solving an optimization problem that considers the statistical properties of the MU spike trains [13]. The theory is closely related to independent component analysis (ICA) [25], which has three main assumptions: the linear superposition of the observations, the prior of the MU spike trains, and the joint prior of the MU spike trains being factorial (i.e., statistical independence). Existing algorithms vary with respect to the selected objective functions (e.g., higher-order statistical moments such as skewness or kurtosis), optimization methods (e.g., gradient descent or quasi-Newton methods), and (partially algorithm-specific) hyperparameters (often in the range of 10-50). Popular CBSS algorithms include the gradient convolution kernel compensation (gCKC) [21], as well as variants of fast Independent Component Analysis (ICA) with [8] or without peeling off (removing) the signal contributions of already identified MU spike trains [37] and reiterating the spike train estimation procedure.

**Deep-learning outlook.** Recent work [35] demonstrates that a shallow autoencoder (AE) with an orthogonally constrained encoder and a sparsity-promoting latent objective can recover spike-like sources from EMG data without labels, representing a deep learning alternative to traditional ICA-based approaches. Other approaches are still in an exploratory phase. Recent *non-linear ICA*

---

[1] https://github.com/dfarinagroup/muniverse

frameworks offer a principled route to overcome the accuracy–latency limits of classical pipelines and, crucially, to deal with non-stationary signals. In particular, time-contrastive learning (TCL) trains a classifier to discriminate short, non-stationary segments of a time series; the resulting latent space provably recovers the independent sources up to trivial indeterminacies [24]. Follow-up work with auxiliary variables [26] and identifiable variational autoencoders [28] generalizes this idea, demonstrating that deep networks can achieve identifiability without ground-truth sources – an essential property for EMG decomposition, where verified MU labels typically do not exist, and if known, might change over time. Although these methods have been explored in modalities such as EEG, they have yet to be brought to EMG. Integrating such self-supervised objectives with architectures already validated for *supervised* HDsEMG decomposition [9, 31, 43] could yield low-latency, device-independent systems that learn *on-the-fly* from raw recordings – eliminating the whitening and delay-embedding steps that hamper current approaches and opening the path to robust MU decoding during natural movements.

### 2.3 Public EMG datasets, algorithms and benchmarking efforts

Over the past few years, a handful of open-source toolboxes [3, 39, 42, 27] have begun to lower the barrier to EMG decomposition research. However, most studies have been based on in-house codes or proprietary packages. Experimental data releases [6, 2, 27] have become more frequent in recent years; however, standards for reporting both data and metadata are still lacking, hindering their integration into fully automated pipelines. Other sEMG collections [1, 11, 40, 41] address classification or myoelectric control tasks but lack ground truth in terms of spike trains.

In contrast, fields such as spike sorting (through efforts like SpikeForest [34]) and EEG source localization (through BCI Competitions) have access to large, standardized benchmarks that drive rapid progress. However, EMG decomposition remains fragmented across private data, divergent pipelines, and inconsistent reporting, motivating the need for a unified, extensible benchmark.

## 3 MUniverse overview

### 3.1 Design goals

We designed MUniverse to be modular, reproducible, and accessible to both neuroscientists and machine learning researchers. For this purpose, there are three goals for MUniverse:

- **FAIR and BIDS compliance:** to facilitate users in rapidly finding and accessing the datasets, we provide Croissant files for each curated dataset. Each dataset adheres to the recently released BIDS-EMG standard [2]. In short, BIDS (Brain Imaging Data Structure [17]) is a community standard for organizing neuroimaging data and metadata in both human- and machine-readable formats, and it comes with existing pipelines to process and ingest datasets stored in this format.

- **Rich metadata and provenance:** to ensure that the various datasets and results we provide and operate on remain end-to-end reusable and reproducible, we provide automated logging to carry full provenance on raw signals, simulation/algorithm parameters, and decomposition outputs.

- **Modularity and containerization:** to isolate dependencies and ensure cross-platform reproducibility, we provide a clean and modular API for data generation, decomposition, as well as evaluation, while each of the external packages (NeuroMotion [33] and Swarm-Contrastive Decomposition (SCD) [19]) is containerized.

### 3.2 High-level architecture

MUniverse is organized into four high-level modules: (1) the simulation stack, (2) decomposition pipelines, (3) evaluation and benchmarking tools, and (4) utilities to handle all data formatting and reporting, enabling users to produce and ingest data and metadata seamlessly (Figure 2).

- **Simulation stack:** MUniverse features a user-friendly API to a state-of-the-art EMG simulator, NeuroMotion [33]. NeuroMotion combines OpenSim-based muscle kinematics, motor neuron-pool models to generate spike trains [16], and a MUAP waveforms generator [32], to synthesize high-fidelity data that closely matches the biophysics of EMG generation. Through MUniverse, users can supply a config file specifying the anatomical, movement, and recording parameters that govern EMG. The simulator module parses this config to invoke NeuroMotion under the hood and emit raw EMG together with ground-truth spike trains, kinematic variables of the

---

[2]https://bids-specification.readthedocs.io/en/latest/

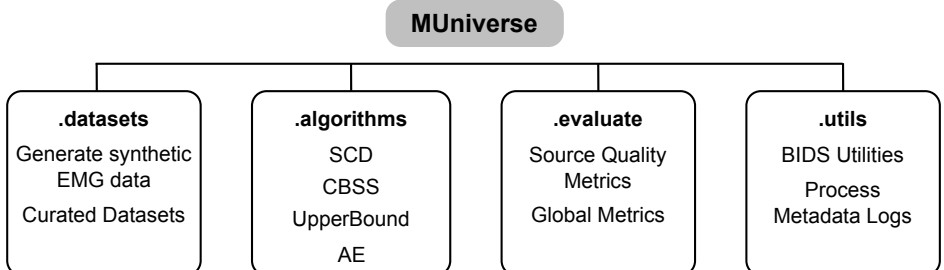

Figure 2: **MUniverse simulation and benchmarking suite**: MUniverse is organized into four high-level modules, providing a user-friendly API to (1) load and simulate data, (2) decompose HDsEMG recordings, (3) evaluate performance and (4) organize data into a standard format.

degree of freedom under consideration, torque/activation profile that generated the spike trains, and a complete JSON provenance record.

- **Decomposition routines:** the framework implements a uniform API to four decomposition methods - (1) a well-established CBSS algorithm based on FastICA [37], (2) a containerized version of the SCD algorithm [19], (3) a deep autoencoder-based decomposition algorithm [35], and (4) a linear upper-bound method that is only available for simulated data [29, 30].

- **Evaluation and benchmarking:** we provide convenience functions to evaluate decomposition algorithms and generate report cards of algorithm performance. We implemented several common metrics to evaluate both the quality of the estimated MU spike trains (individually) and the overall reconstruction accuracy.

- **Data standardisation and logging:** We offer tools to organise experimental HDsEMG datasets into BIDS structure for interoperability with existing tools like MNE [18] that operate on this standardised format.

### 3.3 Usage and Access

The MUniverse codebase is publicly available on GitHub under a GNU-GPL3 license, providing researchers with complete access to all components. Accompanying datasets are hosted on Harvard Dataverse [3] with comprehensive documentation and made available through Croissant files, enabling efficient integration with existing workflows and analytical pipelines. This open-access approach encourages community-driven improvements to the framework.

## 4 Data generation and curation

### 4.1 Datasets overview

Our dataset collection covers muscles with different sizes and architectural properties (e.g., tibialis anterior and forearm muscles), isometric (slow and ballistic) and dynamic movements, and electrode arrays with different densities (Table 1), thereby rigorously challenging decomposition algorithms under realistic variability. We curated three complementary types of HDsEMG datasets – Synthetic, Hybrid, and Experimental – to span a wide spectrum of muscles, contraction types, recording configurations, and noise conditions.

1. **Synthetic:** Generated end-to-end by the NeuroMotion pipeline [33]. Here, a user-defined 'movement profile' (e.g., a trapezoidal isometric contraction of the extensor carpi muscle at 50% MVC) drives a motor neuron-pool model to produce spike trains [16], which are then convolved with a virtual subject's specific MUAPs. We generated two synthetic datasets – NeuroMotion Train set and NeuroMotion Test set, of 10,000 and 985 recordings, respectively.

2. **Hybrid:** Uses the same pipeline as in the synthetic datasets, but replaces synthetic MUAPs with experimentally recorded MUAPs (see paragraph "Experimental MUAP library" below), preserving fully simulated spike trains while testing algorithm robustness given real MUAP waveforms. Here, we generated a dataset of 100 recordings of isometric contractions.

---

[3]https://dataverse.harvard.edu/dataverse/muniverse

3. **Experimental:** Open access HDsEMG recordings from the tibialis anterior or vastus lateralis muscle during trapezoidal isometric contractions of variable intensity (Caillet et al. (2023) [7]; Avrillon et al. (2024) [4]; Grison et al. (2025) [19]). For Grison et al. (2025), each file is accompanied by MU spike train estimates extracted in vivo from synchronised high-density intramuscular EMG, providing an independent, high-precision reference for benchmarking.

Table 1: MUniverse provides a curated library of datasets that span a wide spectrum of muscles, contraction types, recording configurations, and noise conditions.

| Dataset Name | N. Recordings | N. Muscles | Effort (%MVC) | Movement Profiles |
|---|---|---|---|---|
| NeuroMotion Train | 10,000 | 7 | [10, 100] | 6 |
| NeuroMotion Test | 985 | 7 | [10, 80] | 6 |
| Hybrid Tibialis | 100 | 1 | [10:5:100] | 2 |
| Caillet et al. (2023) | 11 | 1 | {30, 50} | 1 |
| Avrillon et al. (2024) | 124 | 2 | [10:10:70] | 1 |
| Grison et al. (2025) | 10 | 1 | [10:10:70] | 1 |

## 4.2 Simulated datasets: synthetic and hybrid

Our synthetic and hybrid datasets were generated through a principled experimental design framework. We employed Latin hypercube sampling (LHS) to efficiently explore the high-dimensional parameter space of muscle types, movement profiles, effort levels, and recording configurations. This approach ensures comprehensive coverage of possible EMG signals while minimizing dataset size and computational overhead.

**Dataset properties and variants** For the synthetic dataset generation pipeline, we first defined the parameter space spanning all relevant factors listed below. We then used LHS to create 10,000 parameter combinations for the train set and 985 for the test set, ensuring balanced representation across the parameter space. Each combination invoked NeuroMotion to generate the recording, while our package enabled reproducible, scalable, and distributed data generation.

- **Target Muscles:** 7 forearm muscles that control the flexion-extension and radial-ulnar deviation of the wrist; tibialis anterior muscle for the hybrid dataset.
- **Movement types:** isometric contractions, dynamic flexion-extension and radial-ulnar deviation movements.
- **Movement profiles:** trapezoid (ramp and hold), triangular, sinusoid, and ballistic contractions. The degree of freedom (DoF) movement angle was varied for dynamic contractions, while effort in terms of maximal voluntary contraction (MVC) was varied for isometric contractions.
- **Effort ranges:** 5 – 80 % MVC.
- **SNR levels:** 10 – 30 dB additive Gaussian noise.
- **Electrode configurations:** grids (10x5, 10x10) and bracelet (10x32).

For the hybrid dataset, we followed a similar approach but with the additional step of collating experimentally recorded MUAPs.

**Experimental MUAP library:** An experimental MUAP library and the corresponding recruitment thresholds in % MVC were extracted from experimental 256-channel recordings from eight subjects from the dataset Avrillon et al. (2024) [2, 4]. A total of 1031 MUAPs were extracted using spike-triggered averaging with a $\pm 25$ ms window of each of the curated spike trains. To ensure a compact support of the extracted MUAPs due to non-zero edge values, we applied a Tukey window with a cosine fraction of 0.1 [30]. Then, a specified number of the 1031 MUAPs and their recruitment thresholds were randomly sampled for convolving with the generated spike trains for EMG signal generation.

From here, we proceeded to produce 100 recordings by varying the movement profile, effort levels, and noise conditions. Different subsamples of the 1031 MUAPs were obtained to simulate subject-specific parameters.

## 4.3 Experimental HDsEMG datasets

We re-purposed three published datasets (Caillet et al. (2023) [7]; Avrillon et al. (2024) [4]; Grison et al. (2025) [19]) consisting of HDsEMG signals recorded on the Tibialis Anterior muscle or Vastus

Lateralis muscle. All datasets are acquired from published sources, experiments were conducted with informed consent, approved by an institutional review board, and complied with the Declaration of Helsinki. In Caillet et al. (2023) and Avrillon et al. (2024), four 64-electrode grids were used. In these datasets, trapezoidal contractions of randomized order and in the range of 5-80% MVC were performed. In Grison et al. (2025), surface EMG was recorded using two 64-electrode grids, in conjunction with invasive EMG (three multi-channel electrode arrays). Intramuscular and surface EMG signals were concurrently sampled at 10,240 Hz.

### 4.4 Dataset formatting and metadata

All datasets are stored in the open and standardized BIDS format [17] and are rich in metadata to ensure both reusability and interoperability. The datasets include global metadata (e.g., participant population and protocol descriptions) as well as recording-specific metadata (e.g., hardware specifications, channel annotations, and global electrode coordinate systems). The MUniverse package includes routines for easily generating and reading BIDS datasets, facilitating the enrichment of the existing database as well as easy integration into fully automated decomposition pipelines.

## 5 Algorithms and benchmarking

### 5.1 Decomposition algorithms

For all used decomposition approaches, one makes use of the fact that the convolutive mixture in Equation 1 can be reformulated into a linear instantaneous mixture by introducing an extended vector of MU spike trains and observations, each including the original MU spike trains or observations and their delayed versions [13]. After applying a whitening transformation, spike trains can be estimated recursively by applying a projection vector, i.e., representing a column of the inverse extended and whitened mixing matrix, to the extended and whitened signals. In MUniverse, we implemented and evaluated two of the most successful approaches to EMG decomposition (**CBSS** [37] and **SCD** [19]) along with a deep autoencoder-based decomposition (**AE** [35]) and a linear upper-bound algorithm requiring prior knowledge of the MUAPs (**UB**) [30]. We describe an overview of these algorithms, where an extended description is presented in Appendix A.

**FastICA-based decomposition (CBSS):** The CBSS algorithm has 25 hyperparameters, capturing signal pre-processing, signal extension, whitening, separation vector optimization, spike detection, and automated quality classification of the estimated spike train [37]. The optimization problem (Equation (A.3)) is iteratively optimized by a fixed point algorithm (i.e., a quasi-Newton solver with quadratic convergence) with a sparse-based contrast function. Different strategies on optimization algorithm initialization can be selected, as well as strategies in preventing the algorithm from repeatedly converging to the same source (subspace projection methods vs. peeling off MUs from the EMG signal given the estimated spike trains).

**Swarm-Contrastive Decomposition (SCD):** SCD preserves the CBSS backbone but replaces the fixed contrast function with a higher-order cumulant selected on-the-fly via particle-swarm search. It uses a peel-off loop to remove each accepted spike train before the next pass. In addition to the 25 hyperparameters, two extra hyperparameters (swarm size and silhouette threshold) are introduced.

**Deep-learning baseline (AE):** The deep learning approach [35] is fully unsupervised and casts ICA as an autoencoder (AE) problem, where the latent dimension corresponds to the number of MU spike trains. The encoder is an orthogonal rotation that maps extended and whitened observations to latent activations, encouraging separation of different MU spike trains. The decoder is a linear layer followed by tanh-shrink, mapping latents back to the whitened, extended space. Training minimizes a reconstruction term plus a sparsity penalty on the latents.

**Upper-bound (UB):** In a simulated dataset, given the ground truth MUAP waveforms, the upper-bound (UB) accuracy of a linear CBSS algorithm can be calculated by directly computing the projection vector. Due to the extension, there are multiple delayed copies of the same spike train. Therefore, we select the column of the extended and whitened MUAP with the largest $L_2$-norm [30].

### 5.2 Benchmarking workflow

For each recording and pipeline, we executed a standardized workflow: the raw EMG and its JSON sidecar are read into memory; the selected engine produces MU spike train estimates alongside process metadata (e.g., runtime and the selected hyperparameters). For the synthetic and hybrid datasets, this was followed by a spike train-matching procedure that compares the algorithm outputs

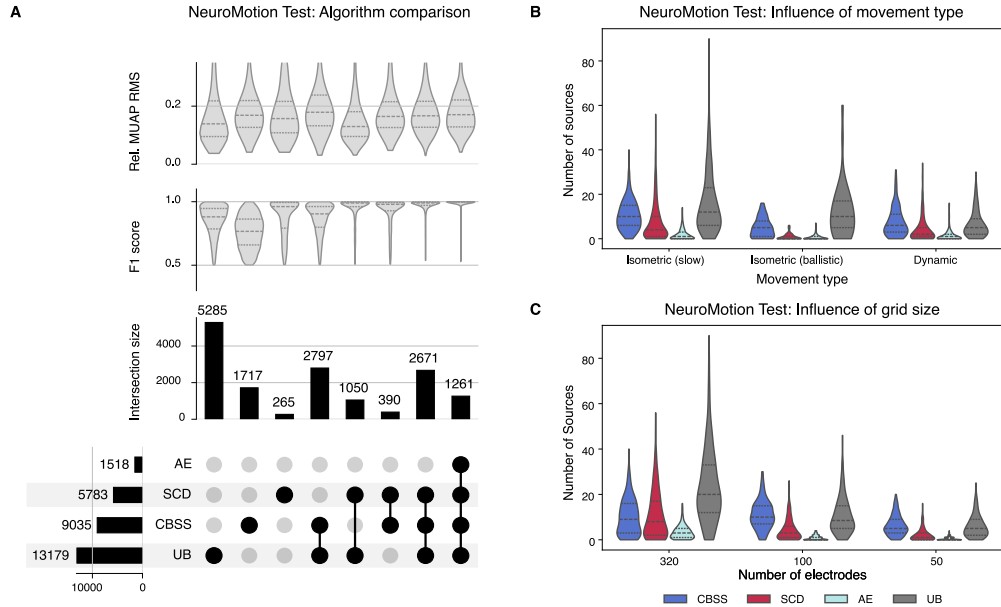

Figure 3: Decomposition performance given the NeuroMotion test dataset, considering all units with an F1 score above 0.5. (A) UpSet plot showing the total number of identified units per algorithm (horizontal bars) and the intersection sizes of units common between different sets of algorithms (vertical bars; only considering sets with a minimum number of 150 units). Further, for each set, the distributions of the F1 score and the relative MUAP amplitudes are shown. Violone plots showing the number of estimated MU spike trains per algorithm (indicated by different colors) depending on the movement type (B) and the number of EMG electrodes (C).

with the reference spike trains. For matching, the fraction of common spikes was calculated for all pairs of predicted and ground truth MU spike trains. Thereby, predicted and ground truth MU spike trains were temporally aligned based on cross-correlation within a ±100 ms window. Then, estimated spike trains were assigned to a ground-truth spike train by solving a linear sum assignment problem with the false positive rate as the cost function and only accepting matches when two spike trains share at least 30 % of common spikes. Finally, the outputs of each pipeline were evaluated using the following metrics to produce the final results.

### 5.3 Evaluation metrics

Each pipeline is evaluated using two sets of metrics. The first set applies across all dataset types and includes (a) source quality metrics such as the silhouette-like score, (b) the number of estimated MU spike trains above the quality thresholds in (a), and (c) the fraction of variance explained (FVE) by the identified and accepted spike trains (see Appendix C). The FVE is computed by recursively peeling off the contribution of each identified spike train. This residual signal lets us directly estimate the variance explained by the identified MU spike trains. The second set of metrics applies to datasets with known ground-truth spike times or expert-annotated decompositions from intramuscular recordings. For these, we computed the number of true positives, false positives, and false negatives (post cross-correlation-based spike train matching, see Appendix C). We report the number of matched units (absolute and relative), along with their F1-scores.

## 6   Results

Here, we showcase the performance of three different algorithms (plus the linear upper bound estimate) on 5 datasets, leading to the first comprehensive benchmarking effort in EMG decomposition.

### 6.1   Global summary and source quality metrics

Table 2 summarizes the decomposition performance on the Avrillon et al. (2024) and Neuromotion Test datasets, which are chosen as representative samples of the experimental and synthetic dataset types, respectively (see Table A.1 for the remaining datasets). We report the 10th, 50th (median), and

90th percentiles for each metric across pipelines, and provide brief comments on the most noteworthy findings below.

- **Unit yield:** For experimental data without ground-truth, the unit yield was defined as the number of units with a silhouette score greater than 0.9 and at least 50 detected spikes. For simulated data, the unit yield was defined as the number of estimated MU spike trains with an F1-score greater than 0.5. In all experimental datasets, SCD exhibits the best top-level performance (as indicated by the 90th percentile yield numbers) and achieves the best median performance in two out of three experimental datasets. On the other hand, CBSS outperforms SCD on the simulated datasets. The AE algorithm performs significantly worse than CBSS and SCD.
- **Explained variance:** Similar to the unit yield performance, SCD slightly outperforms CBSS in median FVE on the experimental datasets, while CBSS performs better on the simulated datasets. At the 90th percentile, SCD leads CBSS across four out of the five datasets.

## 6.2  Evaluation with known ground-truth spike trains

Table 3 shows decomposition performance when a ground-truth is available, i.e., an expert reference decomposition based on invasive EMG (Grision et al. (2025)) or the simulated ground truth (NeuroMotion test, Hybrid Tibialis). We present the total number of predicted units, the fraction of estimated MU spike trains matched with a corresponding ground-truth spike train, and the percentile of the spike train accuracy, quantified by means of the F1 score.

- **Summary:** All tested algorithms reliably reconstruct MU spike trains, indicated by a median F1 score above 0.9. Although CBSS showed the highest unit yield in the simulated datasets, SCD is superior in terms of spike train accuracy (i.e., showing higher F1 score values). The lower fraction of matched MU spike trains for UB (75.8 %) is due to the fact that reconstructing the activity of a full MU pool is often not feasible even if the forward model is known.
- **Insights from simulations:** Figure 3 exemplarily shows for the NeuroMotion test dataset how simulations enable detailed insights into the absolute and relative performance of different decomposition algorithms. We note that a considerable gap still exists between established decomposition methods and the theoretical upper bound (Figure 3A); however, these units are the most challenging, which is represented in the F1 score distribution. Generally, units that are detected by all algorithms show the highest accuracy. Though CBSS shows the highest unit yield, SCD is best in detecting low-amplitude units. Considering different tasks (Figure 3B), for all algorithms, the unit yield and decomposition accuracy decrease for dynamic or ballistic contractions. Moreover, it is observed that though CBSS shows the best median unit yield in slow isometric conditions, SCD shows the highest best-case unit yield. Finally, we note that the number of estimated MU spike trains increases with the number of EMG electrodes (Figure 3C). Interestingly, for the smallest grid size (50 electrodes), the CBSS performance closely matched the upper bound prediction, potentially because it becomes more challenging to adequately explore the solution space as the problem dimension increases.

For further details regarding the selected hyperparameters and an in-depth analysis of the Hybrid Tibialis dataset, see Appendix B.

Table 2: High-level performance metrics across datasets and algorithms. (10 | 50 | 90 percentile)

| Dataset | Algorithm | Unit yield | FVE | Runtime (s) |
| --- | --- | --- | --- | --- |
| Avrillon et al. (2024) | CBSS | 0.0 | 14.5 | 52.7 | 0.07 | 0.34 | 0.61 | 85 | 195 | 288 |
| | SCD | 0.0 | 13.0 | 69.5 | 0.00 | 0.34 | 0.67 | 86 | 348 | 805 |
| | AE | 2.0 | 10.0 | 37.0 | 0.00 | 0.10 | 0.23 | 61 | 88 | 126 |
| NeuroMotion Test | CBSS | 1.0 | 8.0 | 18.0 | 0.07 | 0.27 | 0.46 | 126 | 381 | 1046 |
| | SCD | 0.0 | 3.0 | 16.0 | 0.00 | 0.12 | 0.47 | 0 | 147 | 399 |
| | AE | 0.0 | 1.0 | 4.0 | 0.00 | 0.04 | 0.18 | 0 | 13 | 296 |
| | UB | 2.0 | 9.0 | 32.0 | 0.05 | 0.22 | 0.45 | 4 | 11 | 90 |

## 7  Discussion, limitations and future work

Using the proposed MUniverse framework, we perform a standardised comparison of 3 decomposition pipelines across five synthetic and experimental EMG datasets (1230 recordings), marking the largest

Table 3: Decomposition performance with respect to expert reference decomposition (intramuscular recording in the Grison et al. (2025) dataset) or simulated ground truth. (10 | 50 | 90 percentile)

| Dataset | Algorithm | N. Sources | Matched Sources (%) | F1-score |
|---|---|---|---|---|
| Grison et al. (2025) | CBSS | 272 | 12.1 | 0.74 \| 0.92 \| 0.98 |
| | SCD | 325 | 11.1 | 0.78 \| 0.92 \| 0.99 |
| | AE | 159 | 11.9 | 0.65 \| 0.92 \| 0.99 |
| NeuroMotion Test | CBSS | 11198 | 86.0 | 0.60 \| 0.96 \| 1.00 |
| | SCD | 6648 | 90.1 | 0.79 \| 1.00 \| 1.00 |
| | AE | 1717 | 95.5 | 0.81 \| 0.99 \| 1.0 |
| | UB | 18040 | 75.8 | 0.69 \| 0.93 \| 1.0 |
| Hybrid Tibialis | CBSS | 1420 | 98.4 | 0.69 \| 0.92 \| 0.99 |
| | SCD | 843 | 99.5 | 0.81 \| 0.99 \| 1.00 |
| | AE | 228 | 97.4 | 0.77\| 0.97 \| 0.99 |
| | UB | 4217 | 97.7 | 0.79 \| 0.91 \| 0.98 |

such effort ever. Despite the different optimisation strategies and algorithms, all pipelines showed remarkably consistent median performance across experimental and simulated datasets, encouraging neuroscientists and neurophysiologists using these methods to draw robust inferences. No single decomposition pipeline can be considered optimal for arbitrary recordings. Yet, the presented results provide guidance for users in selecting an appropriate decomposition algorithm for specific applications. For example, CBSS is the most robust general-purpose algorithm, and SCD shows the best top-level performance and is most reliable in fully automated settings. This could motivate the use of ensembles of decomposition algorithms or configurations in applications where runtime is not a significant concern. Notably, the inclusion of a theoretical upper bound demonstrates that even for isometric tasks, existing decomposition methods need to be further improved to lower the gap with respect to the theoretical optimal performance. Further, although the deep-learning baseline can currently not compete with established decomposition pipelines, for the first time, we show that an unsupervised neural network architecture can decompose motor unit activity from HDsEMG.

While MUniverse addresses a critical gap in standardized EMG decomposition benchmarks, several limitations remain that we plan to tackle in future releases. In this paper, we have avoided a hyperparameter search for each recording (which is typically done in practice) and instead on a heuristic hyperparameter search over a subset of the most important hyperparameters to manage compute resources. The estimated spike trains have not been post-processed as is typically done in experimental studies. Further, despite the versatility, our current NeuroMotion integration generates EMG only for a single muscle at the moment. In future versions, we will make the API more flexible to be able to directly take kinematic variables as inputs, instead of defining them in a confined manner. Finally, some widely used packages (e.g., MUEdit, Demuse) could not be included in the benchmark. We invite their maintainers to contribute containerized wrappers.

## 8  Conclusion

We have introduced MUniverse, an open, extensible simulation and benchmarking suite for HDsEMG decomposition that unifies data generation, algorithm interfaces, and evaluation under FAIR principles. By providing both massive synthetic corpora and curated experimental datasets alongside containerized decomposition pipelines and standardized metrics, MUniverse lays the foundation for reproducible comparison and rapid method development. We invite the community to adopt MUniverse for fair benchmarking, to contribute new datasets and algorithms, and to collaboratively drive forward the accuracy and reliability of EMG-based neural interfacing.

## 9  Broader Impact

MUniverse democratizes access to high-fidelity EMG data and transparent evaluation, accelerating neural-engineering research from basic science to assistive technologies. By lowering barriers to method development, it has the potential to improve clinical diagnostics, prosthetic control, and our understanding of motor disorders—but also underscores the responsibility to validate these methods across diverse populations and tasks before clinical deployment.

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

# Appendix

## A    Decomposition algorithms

Many successful BSS methods, such as ICA, consider the inversion of an (often ill-posed) linear system. Such BSS approaches can be applied to the decomposition of EMG signals as a convolutive mixture with finite impulse response filters (see Equation 1) can always be written in terms of a linear instantaneous mixture. This reformulation requires introducing an extended vector of MU spike trains and observations, each including the original MU spike trains or observations and their $R$ delayed versions [13]:

$$\tilde{\mathbf{x}}(t) \;=\; \tilde{\mathbf{H}}\tilde{\mathbf{s}}(t) \;+\; \tilde{\boldsymbol{\varepsilon}}(t) \;. \tag{A.1}$$

The signal extension is followed by a whitening transformation, i.e., $\tilde{\mathbf{z}}(t) = \mathbf{V}\tilde{\mathbf{x}}(t)$ where the (non-unique) whitening matrix $\mathbf{V}$ is constructed such that the covariance matrix of extended and whitened observations $\tilde{\mathbf{z}}(t)$ is the identity matrix. This transformation approximately orthogonalizes the mixing matrix and hence, spike trains can be estimated recursively by applying a projection vector $\mathbf{w}_k^T$ to the extended and whitened multichannel EMG signals $\tilde{\mathbf{z}}(t)$:

$$\widehat{s}_k(t) \;=\; \mathbf{w}_k^T \tilde{\mathbf{z}}(t) \;. \tag{A.2}$$

Therein $\widehat{s}_k(t)$ is a non-unique, arbitrarily scaled, and potentially delayed estimate of the $k$th MU spike train. Different variants of convolutive BSS vary regarding the methods used to estimate the columns $\mathbf{w}_k$ of the inverse mixing matrix.

**FastICA-based decomposition (CBSS):**    MUniverse contains an implementation of a state-of-the-art convolutive BSS (CBSS) algorithm based on FastICA; for details, see [37]. The algorithm uses 25 hyperparameters, capturing signal pre-processing, signal extension, whitening, separation vector optimization, spike detection, as well as automated quality classification of the estimated spike train. In contrast to existing implementations, the objective function is

$$\mathcal{L}^{\mathrm{cbss}}(\mathbf{w}_k) \;=\; \sum_t \mathbb{E}\Big(G(\mathbf{w}_k^T \tilde{\mathbf{z}}(t))\Big) \;, \qquad G(s) \;=\; s(s^2 + \epsilon)^{\frac{a-1}{2}} \;, \tag{A.3}$$

where $\epsilon = 0.001$. Adjusting the parameter $a > 0$ allows for fine-tuning the degree of non-linearity based on a simple continuous and infinitely differentiable function. Notably, selecting $a = 3$ is (approximately) equivalent to using skewness as the objective function. The optimization problem posed by Equation (A.3) is iteratively optimized by a fixed point algorithm (i.e., a quasi-Newton solver with quadratic convergence), and different initialization strategies can be selected (random weights or activity index [21]). Moreover, the user can select between different strategies in preventing the algorithm from repeatedly converging to the same source (i.e., different subspace projection methods as well as peeling off MUs from the EMG signal given the estimated spike trains).

**Swarm-Contrastive Decomposition (SCD):**    MUniverse integrates Swarm-Contrastive Decomposition, which preserves the ICA backbone but replaces the fixed contrast with a *source-specific* higher-order cumulant selected on-the-fly via particle-swarm search. For each projection vector $\mathbf{w}_k$ it maximises

$$\mathcal{L}^{\mathrm{scd}}(\mathbf{w}_k) \;=\; \sum_t \mathbb{E}\Big(G(\mathbf{w}_k^T \tilde{\mathbf{z}}(t))\Big) \;, \qquad G(s) \;=\; \mathrm{sign}(s)\,|s|^a \;, \tag{A.4}$$

where the exponent $a$ is optimized, using particle swarm optimization, for every candidate spike train estimate, and a peel-off loop removes each accepted spike train before the next pass. Further, $\mathrm{sgn}(s)$ denotes the signum function. Two extra hyperparameters (swarm size and silhouette threshold) are introduced, allowing SCD to slot into existing frameworks.

**Deep-learning baseline (AE):**    As a deep-learning baseline, MUniverse includes an implementation of the autoencoder approach [35]. This method casts ICA as an autoencoder problem, where the latent dimension corresponds to the number of sources. The encoder is an orthogonal rotation $V \in \mathrm{SO}(mR)$ that maps extended-whitened observations to latent activations, encouraging separation of different sources. The decoder is a linear layer followed by tanhshrink, mapping latents back to the observation space. Training minimizes a reconstruction term plus a sparsity penalty on the latents:

$$\mathcal{L} = \|\tilde{\mathbf{x}}_w - \hat{\mathbf{x}}\|_2^2 \;+\; \lambda \, \log_{10}\left(\frac{q}{p}\,\frac{\|\mathbf{s}\|_p}{\|\mathbf{s}\|_q}\right), \qquad 0 < p < q. \tag{A.5}$$

The method is fully unsupervised, aligns with the classical "rotation-after-whitening" view from ICA, and applies to both iEMG and HD-sEMG. After training, a simple peak finding algorithm is used on the latents to produce the estimated sources.

**Upper-bound (UB):**  For robust learning, the projection vector $\mathbf{w}_k^T$ has been shown to converge to a scaled version of the extended and whitened MUAP [30]. Therefore, in a simulated dataset, the upper-bound accuracy of a CBSS algorithm can be calculated by directly computing the projection vector given the ground truth MUAP waveforms. In detail, the spike train of MU $k$ with delay $l$ is

$$\widehat{\widetilde{\mathbf{s}}}_{k,l}(t) \;=\; \frac{\widetilde{\mathbf{h}}_{k,l}^T}{||\widetilde{\mathbf{h}}_{k,l}||}\,\widetilde{\mathbf{z}}(t)\;, \tag{A.6}$$

where $\widetilde{\mathbf{h}}_{k,l}$ is a single column of the extended and whitened mixing matrix associated with MU $k$ and $||*||$ denotes the $L_2$-norm. Due to the extension, there are multiple delayed copies of the same spike train. Hence, as for the upper-bound estimate, we maximized the expected spike train amplitude by selecting the column of the extended and whitened MUAP with the largest $L_2$-norm [30].

**Spike train estimation**  Finally, binary spike trains need to be derived from the estimated MU spike trains $\widehat{s}_k(t)$. Therefore, an asymmetric power function, i.e., $G(s) \;=\; \text{sgn}(s) \cdot |s|^a$ is used to enhance the contrast between the MU spikes and background peaks, where the exponent $a \in \mathbb{R}$ is an adjustable parameter (here $a = 2$). The tallest peaks with a minimal distance of 10 ms were extracted, and K-means clustering with $K = 2$ was used to separate the peak heights into two clusters, where the putative MU spikes are those in the cluster with the largest centroid. Based on these clusters, a silhouette-based score was computed as a quality metric for the estimated spike trains [37].

# B   Extended results

Here, we present an extended analysis of the performance of the algorithms on the Hybrid tibialis dataset. We analyze both global and source-level properties of the decomposition results, and compare their relative performances on recordings of varying difficulty. Further, we present results of our hyperparameter optimisation experiments to quantify how a subset of the most important hyperparameters affect the decomposition performance.

## B.1   Insights from simulations

The Hybrid Tibialis dataset strongly mimics experimental conditions that have been extensively used to validate and optimize existing decomposition pipelines. Thereby, all tested algorithms show robust spike train reconstructions, indicated by a median F1 score larger than 0.9 (Table 3). Nevertheless, Figure A.1A shows that existing decomposition pipelines have not reached the theoretical optimal performance, indicated by 2580 units uniquely detected by UB. While the CBSS algorithm is currently the closest competitor, the SCD algorithm is more refined in identifying low-amplitude units. Furthermore, since the SCD predictions yield the highest F1 score, this highlights the method's benefits in fully automated settings (i.e., without human-in-the-loop post-processing). Further, the presented data shows a strong decrease in the decomposition performance for contraction intensities above 50 % of the maximum voluntary contraction (MVC) of the muscle. (Figure A.1B). Notably, for contraction intensities above 50 % MVC, CBSS performs both with respect to median and top-level performance. Although UB shows a higher unit yield in triangular movement profiles than in trapezoidal movement profiles, all decomposition algorithms exhibit similar performance, independent of the movement profile (Figure A.1C).

## B.2   Influence of hyperparameters

The performance of decomposition algorithms critically depends on the selected hyperparameters. Achieving an optimal decomposition performance often requires a per-recording hyperparameter optimization. Although hyperparameter optimisation is beyond the scope of the proposed manuscript, in this section, we exemplify the influence of two hyperparameters on the performance of the CBSS decomposition algorithm. That is, the extension factor $R$ and the strategy for avoiding finding multiple copies of the same source (subspace projection using Gram-Schmidt orthogonalization versus estimating single-source signal contributions and subsequent peel-off). For this purpose, we quasi-randomly selected a subset of 50 recordings from the NeuroMotion test dataset and decomposed it using 6 sets of hyperparameters (i.e., $R = 5, 10, 20$ and using either peel-off or subspace projection). It can be observed that the worst set of hyperparameters (i.e., $R = 5$ and using subspace projection)

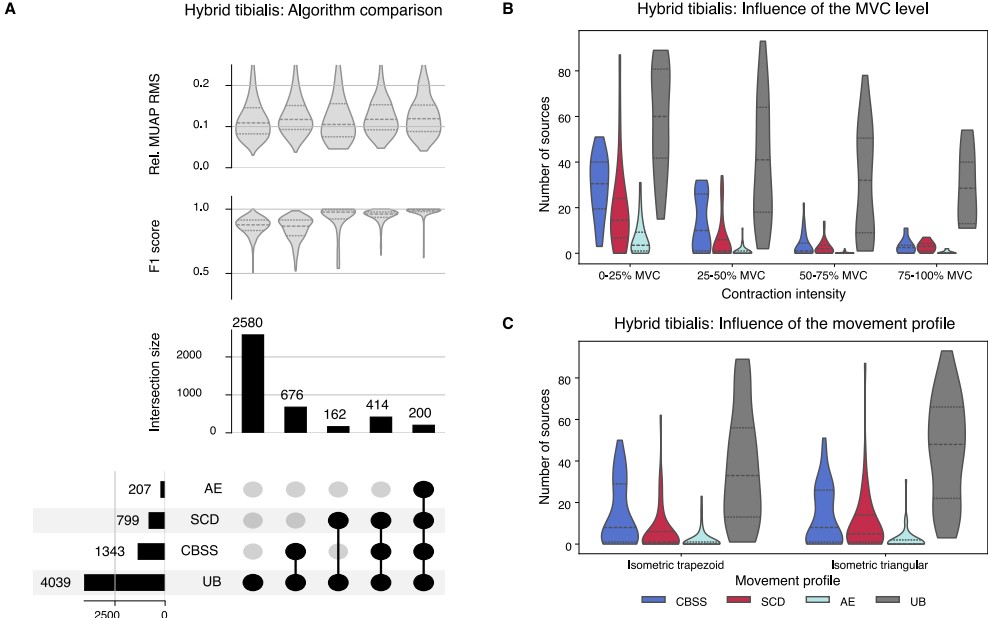

Figure A.1: Decomposition performance on the Hybrid tibialis dataset considering all units with a F1 score above 0.5. (A) UpSet plots showing the total number of detected units per algorithm (horizontal bars) and the intersection sizes of units common between different sets of algorithms (vertical bars; only considering sets with a minimum of 50 units). Violone plots showing the number of sources per algorithm (indicated by different colors) depending on the contraction intensity (B) and the effort profile (C).

Table A.1: High-level performance metrics across datasets and algorithms. (10 | 50 | 90 percentile). Bold font indicates the best-performing algorithm in terms of median yield; the UB algorithm is highlighted in gray to indicate that it is a linear upper-bound estimate.

| Dataset | Algorithm | Unit yield | FVE | Runtime (s) |
|---|---|---|---|---|
| Caillet et al. (2023) | CBSS | 4.0 \| 32.0 \| 48.0 | 0.33 \| 0.51 \| 0.59 | 194 \| 254 \| 308 |
| | SCD | 20.0 \| 38.0 \| 60.0 | 0.39 \| 0.57 \| 0.61 | 314 \| 504 \| 889 |
| | AE | 9.0 \| 15.0 \| 24.0 | 0.17 \| 0.21 \| 0.24 | 77 \| 99 \| 120 |
| Grison et al. (2025) | CBSS | 24.9 \| 25.5 \| 30.0 | 0.21 \| 0.27 \| 0.37 | 574 \| 888 \| 1514 |
| | SCD | 27.0 \| 32.5 \| 34.5 | 0.22 \| 0.32 \| 0.40 | 545 \| 802 \| 1206 |
| | AE | 8.9 \| 10.0 \| 11.1 | 0.13 \| 0.18 \| 0.20 | 87 \| 162 \| 186 |
| Hybrid Tibialis | CBSS | 0.0 \| 8.0 \| 35.2 | 0.00 \| 0.15 \| 0.48 | 0 \| 623 \| 952 |
| | SCD | 0.0 \| 3.0 \| 24.0 | 0.00 \| 0.06 \| 0.43 | 0 \| 316 \| 718 |
| | AE | 0.0 \| 0.0 \| 6.0 | 0.00 \| 0.00 \| 0.15 | 0 \| 0 \| 201 |
| | UB | 8.0 \| 41.5 \| 80.3 | 0.11 \| 0.30 \| 0.45 | 50 \| 64 \| 79 |

detects 72.8 % of the MUs detected by the best performing set of hyperparameters ($R = 10$ and using peel-off), see Figure A.2. Further, for both the peel-off and subspace projection-based approaches, the highest MU yield is obtained for an extension factor of $R = 10$. Interestingly, each set of hyperparameters detects unique MUs (i.e., sources that are not detected by other hyperparameter sets). However, these unique sources contain more errors (i.e., false-positive and false-negative spikes), which is indicated by the fact that the corresponding F1-score distributions are shifted towards lower values.

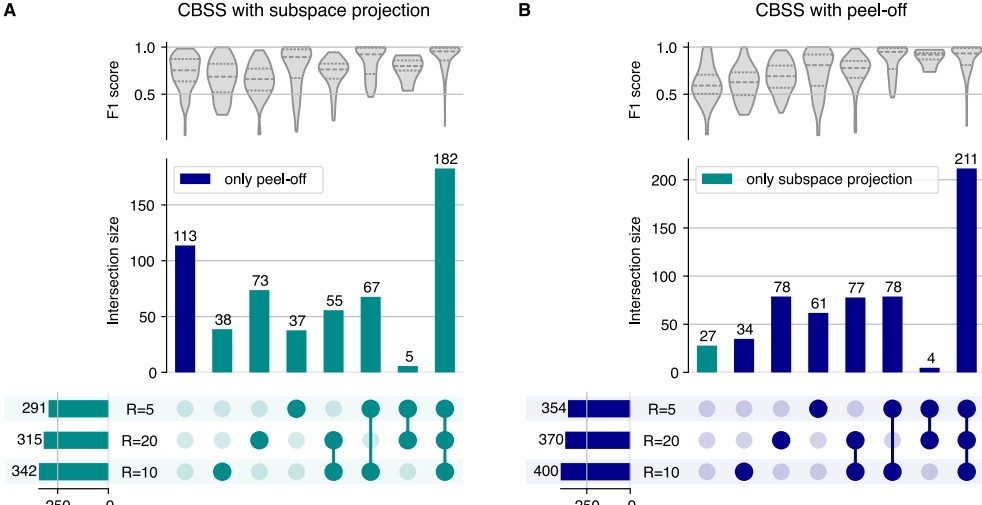

Figure A.2: UpSet plots comparing the performance of different hyper-parameter sets using either subspace-projection (A) or peel-off (B) for avoiding repeated convergence to the same sources, as well as different extension factors $R$. The horizontal bars indicate the overall number of detected units per hyperparameter set. The vertical bars indicate the intersection size of different hyperparameter sets.

## C  Performance metrics

The MUniverse evaluation module provides a large set of performance metrics that facilitate a standardized and holistic comparison of different decomposition algorithms. While labeled ground truth spikes always exist for simulated data, this is only partially possible for experimental recordings, e.g., simultaneously recording invasive signals together with expert knowledge. Thus, next to spike matching-based metrics (Appendix C.1), MUniverse provides several purely signal-based source quality estimates (Appendix C.2).

### C.1  Spike-based quality metrics

First, the predicted and ground-truth spike trains are matched by computing for each pair of motor neuron discharges the fraction of common spikes. To do so, each pair of spike trains is aligned in the time domain by computing the lag that maximizes the cross-correlation function. All pairs of spikes with a maximal delay of $\pm 1$ ms are considered common spikes. We label the predicted spike trains based on the highest fraction of common spikes, whereby it is required that two spike trains corresponding to the same MU have a minimum of 30 % common spikes. Next, we compute the total number of matched spike trains as well as the fraction of matched spike trains, i.e., the total number of matched spike trains divided by the total number of ground truth spikes. Further insights into the decomposition performance are obtained by computing the rate-of-agreement (RoA), precision, recall, and the F1-score:

$$\text{RoA} \;=\; \frac{\text{TP}}{\text{TP} + \text{FP} + \text{FN}} \,, \tag{A.7a}$$

$$\text{Precision} \;=\; \frac{\text{TP}}{\text{TP} + \text{FP}} \,, \tag{A.7b}$$

$$\text{Recall} \;=\; \frac{\text{TP}}{\text{TP} + \text{FN}} \,, \tag{A.7c}$$

$$\text{F1-score} \;=\; 2 \cdot \frac{\text{Precision} \cdot \text{Recall}}{\text{Precision} + \text{Recall}} \,. \tag{A.7d}$$

Therein, TP deontes the number of true positive spikes, i.e., spikes that appear in both spike trains with a maximum delay of $\pm 1$ ms, FP are is the number of false positive spikes that are only part of the predicted spike train, and FN is the number of false positive spikes that are only part of the ground truth spike train.

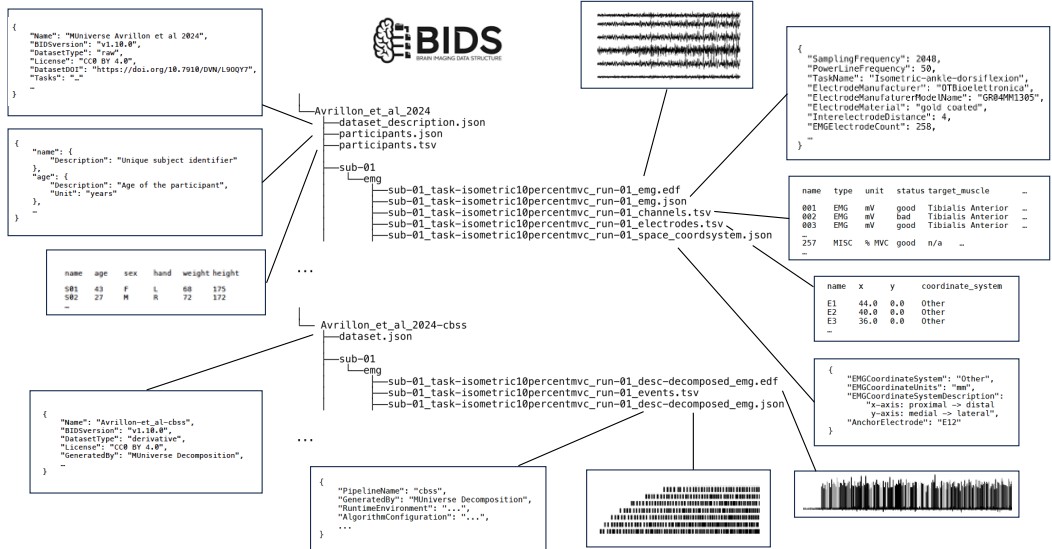

Figure A.3: **Overview of the BIDS specification.** Exemplary illustration of the BIDS folder structure and files used for storing data and metadata in a simultaneously human and machine-readable format. The upper part showcases the BIDS-EMG specification, and the lower part illustrates how decomposition outputs are formatted as BIDS-derivatives.

## C.2 Signal-based quality metrics

To quantify the uncertainty of the predicted sources, we consider a set of quality metrics that rely solely on measurable quantities (i.e., the EMG signals) and the predicted sources. First, we compute for each set of predicted spike trains the variance of the EMG signal that can be explained by the linear convolutive mixture model described in Equation (1). This is achieved by estimating for each predicted MU source the impulse response using spike-triggered averaging (in a window of $\pm 50$ ms around the spike times), and peeling of the MU contribution obtained by convolving the estimated impulse response with the predicted spike trains. The fraction of explained variance (FVE) is given by

$$\text{FVE} = 1 - \frac{\text{Var}\left(\mathbf{x}^{\text{res}}(t)\right)}{\text{Var}\left(\mathbf{x}(t)\right)}, \tag{A.8}$$

where $\mathbf{x}^{\text{res}}(t)$ is the residual signal obtained after peeling off every predicted source. Purely source-based source quality metrics have been proven useful; however, all metrics face limitations. Thus, MUniverse allows users to simultaneously analyse multiple metrics, and we compute for each source the silhouette score [37], the pulse-to-noise ratio [20], the relative peak separation [30], skewness, kurtosis, the (z-scored) mean peak amplitude, and the mean discharge rate, as well as the coefficient of variation of the interspike intervals.

## D  MUniverse Datasets

### D.1  FAIR principles

MUniverse is built on the premise that the EMG research community can hugely benefit from shared, rigorously standardised, and fully reproducible resources. By bringing various types of HD-EMG signals into a single BIDS-compatible library tied to a transparent evaluation suite, it invites community-driven extension.

**FAIR compliance:**

- **Findable (F):** Each dataset is issued a Harvard Dataverse DOI and a persistent URL; the collection is further described by a machine-readable Croissant JSON-LD file that indexes every file within the dataset with a version tag and a SHA-256 checksum.

- **Accessible (A):** Harvard Dataverse's open-source platform serves the data servers over HTTPS and via several Python/ HTTPS endpoints, so both humans and automated agents can fetch the data without login, paywall, or proprietary tooling.
- **Interoperable (I):** All files follow the BIDS (EMG extension) schema (also see Section D.2) and are encapsulated in a Croissant JSON-LD, so open-source tools like the BIDS-validator or MNE-Python can parse them out-of-the-box.
- **Reusable (R):** Signals are licensed under CC-BY-4.0, and the entire simulation / preprocessing pipeline will be made openly accessible via an MIT license. Due to the rich provenance (simulation config JSONs, container digest, environment variables, etc.) that travels with every recording, MUniverse promotes reusability of both the datasets and code.

## D.2 BIDS specification

BIDS is a standardized format for organizing and sharing anatomical and physiological data, to facilitate reproducibility, interoperability, data sharing and the use of automated data processing pipelines. Every BIDS compatible dataset uses a standardized folder structure and naming conventions (see Figure A.3) that are unambiguously related to essential recording metadata such as subject identifiers, the applied protocols and the signal modality. Besides storing the actual data in standardized formats (e.g., EDF), BIDS datasets are rich in metadata stored in JSON or TSV files, which are both human and machine-readable. While BIDS was originally developed for brain imaging data, it has proven to be flexible in capturing diverse anatomical or physiological data, as well as processed data referred to as derivatives. BIDS-EMG (available since version 1.10.2) is currently the only standardized format for reporting EMG data, and all MUniverse datasets utilize this standard. Further, all decomposition results are formatted according to the specifications of BIDS derivatives. The MUniverse utility module contains all essential functionalities for loading existing BIDS-EMG datasets and exporting new datasets in the BIDS-EMG format.

# E  Author contributions and funding

| | PM* | TK* | DH* | AG | IMG | SM | AC | SA | RR | DF |
|---|---|---|---|---|---|---|---|---|---|---|
| **Conceptualization** | ✓ | ✓ | ✓ | | | | | | ✓ | |
| **Methodology** | ✓ | ✓ | ✓ | | | | | | ✓ | |
| **Software** | ✓ | ✓ | ✓ | ✓ | ✓ | | | | | |
| **Validation** | ✓ | ✓ | ✓ | | ✓ | | | | | |
| **Formal Analysis** | ✓ | ✓ | ✓ | | | | | | ✓ | |
| **Investigation** | ✓ | ✓ | ✓ | | | | | | | |
| **Resources – Simulator code** | | | | | | ✓ | ✓ | | | |
| **Resources – Algorithm code** | | ✓ | ✓ | ✓ | | | | | | |
| **Resources – Source data** | | | | ✓ | | | ✓ | ✓ | | |
| **Data Curation** | ✓ | ✓ | | ✓ | | | ✓ | ✓ | ✓ | |
| **Writing – Original draft** | ✓ | ✓ | ✓ | | | | | | ✓ | |
| **Writing – Review & editing** | ✓ | ✓ | ✓ | ✓ | ✓ | ✓ | ✓ | ✓ | ✓ | ✓ |
| **Visualization** | ✓ | ✓ | | | | | | | | |
| **Supervision** | | | | | | | | | | ✓ |
| **Project administration** | ✓ | | | | | | | | | |
| **Funding acquisition** | ✓ | ✓ | | | ✓ | | | | ✓ | ✓ |

\* These authors contributed equally to this work.

**P.M. and I.M.G.** were supported by the Eric and Wendy Schmidt Postdoctoral Fellowship in AI for Science at the I-X Center for AI in Science, Imperial College London. **T.K.** was supported by the German Research Foundation (DFG, Deutsche Forschungsgemeinschaft) through the priority program SPP 2311 (Grant ID: 548605919) and the European Research Council (ERC) through ERC-AdG 'qMOTION' (Grant ID: 101055186). **D. H.** was supported by the Imperial-META Wearable Neural Interfaces Research Centre and the Onassis Foundation under Scholarship ID: F ZT 012-1/2023-2024. **R.R.** was supported by UK Research and Innovation (UKRI) under the UK government's Horizon Europe Guarantee (Grant ID: EP/Z002184/1) and the Swedish Brain Foundation (Grant ID: PS2022-0021).

