# Supplementary Material for "MUniverse: A Simulation and Benchmarking Suite for Motor Unit Decomposition"

**Pranav Mamidanna**[1,2,*], **Thomas Klotz**[3,*], **Dimitrios Halatsis**[4,*],
**Agnese Grison**[2], **Irene Mendez-Guerra**[1,2], **Shihan Ma**[5], **Arnault H. Caillet**[6],
**Simon Avrillon**[7], **Robin Rohlén**[2], **Dario Farina**[2].

[1] I-X Center for AI in Science, Imperial College London, UK
[2] Department of Bioengineering, Imperial College London, UK
[3] Institute for Modelling and Simulation of Biomechanical Systems, University of Stuttgart, Germany
[4] Université Côte d'Azur, LAMHESS, Nice, France
[5] Department of Diagnostics and Intervention, Umeå University, Sweden
[*] These author have equally contributed to the work

## A  Datasheet for MUniverse

### 1. Motivation

**For what purpose was the dataset collection created?** The MUniverse project was developed to address the lack of standardized benchmarks for evaluating MU decomposition algorithms from high-density electromyography (HD-EMG) recordings. By providing a curated, diverse, and extensible suite of datasets with known ground truth (where available), MUniverse enables reproducible comparisons, robust performance assessments, and the development of next-generation decomposition and inference algorithms. (Data curators and funding information will be available upon review.)

**Who created the collection (team / institution)?** Developed by an interdisciplinary team of biomedical engineers, neuroscientists and machine-learning researchers at Imperial College London, University of Stuttgart, Université Côte d'Azur and UmeåUniversity. Full author list as follows: Pranav Mamidanna, Thomas Klotz, Dimitrios Halatsis, Agnese Grison, Irene Mendez-Guerra, Shihan Ma, Arnault H. Caillet, Simon Avrillon, Robin Rohlén, and Dario Farina.

**Who funded the creation of the collection?** Supported by the following grants.

- Eric and Wendy Schmidt Postdoctoral Fellowship in AI for Science. (P.M.)
- Deutsche Forschungsgemeinschaft (DFG, German Research Foundation) through the priority program SPP 2311 (Grant ID: 548605919) (T.K.)
- European Research Council (ERC) through the ERC-AdG 'qMOTION' (Grant ID: 101055186) (T.K.)
- Imperial-META Wearable Neural Interfaces Research Centre and the Onassis Foundation under Scholarship ID: F ZT 012-1/2023-2024. (D.H.)
- UK Research and Innovation (UKRI) under the UK government's Horizon Europe Guarantee (Grant ID: EP/Z002184/1) (R.R.)
- Swedish Brain Foundation (Grant ID: PS2022-0021). (R.R.)

### 2. Composition

**What do the datasets represent?** Datasets represent a large variety of surface EMG recordings spanning multiple muscles, movements, and hardware configurations across three complementary

modalities: *(i) Synthetic (NeuroMotion)* – full biophysical simulations with exact MU spike ground truth; *(ii) Hybrid (Tibialis Anterior)* – experimentally observed MUAP waveforms convolved with simulated spikes, blending realism with perfect labels; *(iii) Experimental* – HD-EMG recordings (Caillet 2023 [3, 4], Avrillon 2024 [1, 2], and Grison 2025 [7]) for robustness tests on real-world data.

**How many datasets are included?** Six datasets, **11 230 recordings** in total: *NeuroMotion-Train* (10 000), *NeuroMotion-Test* (985), *Hybrid-Tibialis* (100), *Caillet 2023* (11), *Avrillon 2024* (124), *Grison 2025* (10).

**What data does each dataset contain?** HD-EMG signals (in binary EDF format), Brain Imaging Data Structure (BIDS) specified sidecars (JSON / TSV) and, when available, ground-truth spike trains (flat TSV format). Synthetic and hybrid sets also store full simulator configurations and provenance logs.

**Are labels or ground truths available?** Yes, for all synthetic recordings and hybrid recordings (exact MU spikes). No labels for experimental sets, except Grison 2025 [7] that includes an intramuscular decomposition result for two-source validation.

**Are any data fields missing?** All required BIDS fields are present; no known omissions.

**Relationships between datasets?** Hybrid MUAPs originate from *Avrillon 2024*; *NeuroMotion-Test* mirrors the train distribution but is held out; the experimental MUAP library underpins the hybrid set.

**Recommended data splits?** *Training* – NeuroMotion-Train. *Held-out test* – NeuroMotion-Test. Experimental and Hybrid-Tibialis datasets are evaluation-only.

**Known noise, errors or redundancies?** Experimental recordings include measurement noise; synthetic sets vary additive Gaussian noise from 10–30 dB; no duplicate files detected.

**Self-contained or external resources?** All raw data are packaged with the release. Simulations rely on the *NeuroMotion* container (included).

**Sensitive or confidential information?** Volunteers are de-identified; no personal health information stored.

## 3. Collection Process

**How were the datasets generated or acquired?** *Synthetic / Hybrid:* generated with a containerized version of the **NeuroMotion** package. *Experimental:* collected with multiple 64-channel HD-EMG grids in approved studies (Caillet et. al. (2023), Avrillon et. al. (2024), Grison et. al. (2025)).

**Who was involved in data creation?** A.H.C. and S.A. were involved in collecting the original experimental data for Caillet et. al. (2023) and Avrillon et. al. (2024). A.G. was involved in collecting the original experimental data for Grison et. al. (2025). D.H., P.M., and I.M.G. were involved in the data creation for NeuroMotion train, NeuroMotion test and Hybrid Tibialis datasets.

**Timeframe of creation?** 2023 – 2025.

**Ethical review processes?** The respective Research Ethics Committees approved each study: Caillet et al. (Imperial College London, reference number 18IC4685), Avrillon et al. (Imperial College London, reference number 18IC4685; Comité de Protection des Personnes Ouest, reference number 23.00453.000166), and Grison et al. (Imperial College London, reference number 19IC5640). The studies followed the Declaration of Helsinki, and the subjects gave informed written consent.

## 4. Pre-processing, Cleaning, Labelling

**Was preprocessing / labelling done?** *Experimental:* hardware band-pass (depending on whether only surface recordings were performed vs hybrid) + 50 Hz notch. *Synthetic / Hybrid:* software 10-500 Hz band-pass filter was applied to all algorithms.

**Is raw data available?** Yes – raw signals are provided for every recording.

**Is the preprocessing / labelling code available?** Yes – MIT-licensed on GitHub `https://github.com/dfarinagroup/muniverse`

**Any other comments?** A JSON provenance record is written automatically for every benchmark run.

## 5. Uses

**Has the dataset been used already?** Yes – for the baseline benchmark in the accompanying MUniverse paper; experimental subsets appear in Caillet et. al. (2023) [3], Avrillon et. al. (2024) [1] and Grison et. al. (2025) [7].

**Repository of related papers / results?** Links to publications and future submissions will be maintained in the project `README`, while provenance of original source material (code, data) is already available through the metadata files accompanying each dataset.

**Suitable tasks?** MU decomposition, simulation-to-real transfer (from kinematic and physiological variables directly into EMG), EMG-to-force modelling, robustness testing under varied noise / artefact profiles.

**Limitations or risks?** Simulations may omit rare artefacts; experimental subject diversity is limited. Experimental datasets retain their own limitations, with lower number of subjects in total.

**Tasks for which the dataset should not be used?** Not validated for direct clinical diagnosis or stimulation-safety decisions.

## 6. Distribution

**Will the dataset be shared publicly?** Yes – via Harvard Dataverse DOIs all available under `https://dataverse.harvard.edu/dataverse/muniverse`, and accompanying Croissant files.

**How will it be distributed?** Data will be distributed through the Harvard Dataverse's persistent URL and DOIs.

**Under what license?** Data under **CC BY 4.0**; code under **MIT**.

**Third-party IP or use restrictions?** None identified.

**Export-control or regulatory limits?** No dual-use concerns; recordings are non-invasive physiological signals.

## 7. Maintenance

**Who maintains the collection?** P.M. and T.K. will continue to maintain the data collection on Dataverse as well as the code on GitHub.

**Contact information?** Any correspondence may be addressed to p.mamidanna@imperial.ac.uk (P.M.), thomas.klotz@imsb.uni-stuttgart.de (T.K.) and d.farina@imperial.ac.uk (D.F.).

**Will updates / errata be provided?** Yes – logged in `CHANGELOG.md` and announced via GitHub releases.

**Will older versions be kept?** Yes – prior DOI-tagged snapshots remain on Dataverse for reproducibility.

**Contribution / extension mechanism?** Pull-requests on GitHub; new datasets accepted after BIDS validation and CI checks; contributors acknowledged in release notes.

**Any other comments?** Community issues and feature requests are tracked through GitHub Discussions.

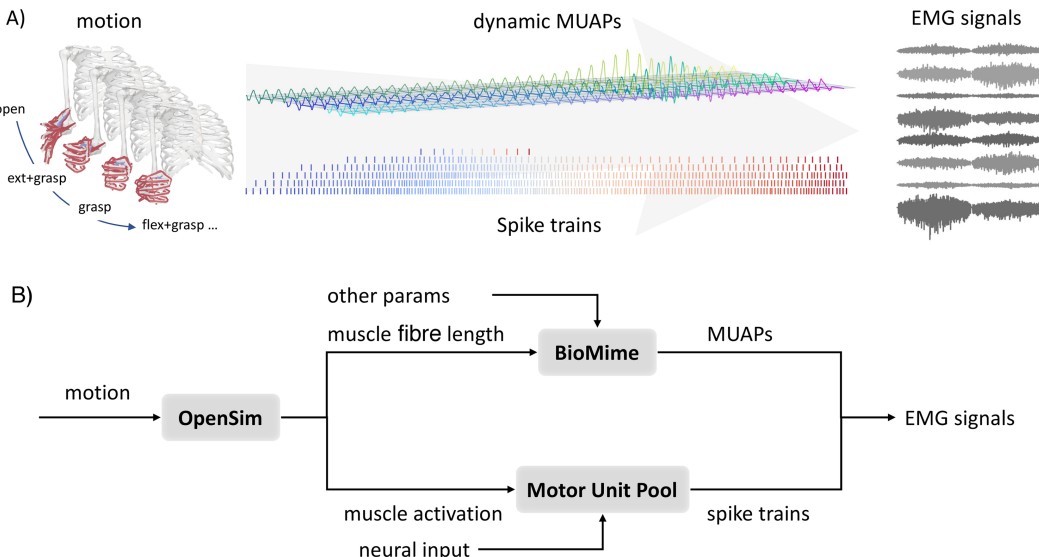

Supplementary Figure 1: Overview of the NeuroMotion pipeline combining BioMime, neural-drive, and musculoskeletal modules for real-time EMG synthesis. Figure reused from [12] with permission from the publisher.

# B   Data generation

## B.1   NeuroMotion EMG simulator

**Context and rationale.** All synthetic surface EMG and ground-truth MU labels in MUniverse were generated with NeuroMotion. The simulator belongs to a broader class of myoelectric digital twins [13] that recreate, in silico, the full chain from the neural drive to the voltages measured at the skin. An anatomically realistic (potentially subject-specific) model begins with 3D imaging (e.g., MRI or ultrasound) from which bone, muscle, fat, and skin surfaces are segmented. Next, the MU pool activity is simulated, which determines the muscle fiber action potentials. In this work, the muscle fiber action potentials are simulated using analytical functions. The electric muscle fiber activities determine the right-hand side of a Poisson problem describing the electric behavior of the body as a volume conductor, and which is approximated numerically, e.g., using the finite element method.

**BioMime: bridging discrete FEM-based simulations to real-time MUAPs**   To avoid having to re-run the finite-element solver at every time frame, NeuroMotion employs BioMime [11], a conditional encoder–decoder model that learns to mimic FEM-derived MUAP templates as a smooth function of six physiological parameters (depth, mediolateral position, fibre length, conduction velocity, innervation-zone location, fibre density). Once trained, BioMime can morph an existing template or sample de-novo from its latent space, allowing NeuroMotion to stream MUAPs in real-time while continuously updating their properties during dynamic contractions.

NeuroMotion fuses the BioMime MUAP generator, a state-of-the-art neural-drive model, and an OpenSim-based musculoskeletal (MSK) model into a modular pipeline that synthesises high-fidelity, HD-EMG signals in real time, even during complex dynamic movements (overview in the Supplementary Figure 1). The individual building blocks are summarized below.

**NeuroMotion modules at a glance**

1. **MSK model (OpenSim ARMs)** – converts joint kinematics (pose sequences or motion-capture trajectories) into time-varying fibre lengths and muscle activations.

2. **Parameter translator** – maps fibre-length changes to conduction-velocity and MU-depth estimates when direct measurements are unavailable.

3. **BioMime generator** – produces time-varying MUAP templates conditioned on the seven physiological parameters.

4. **MU pool** (Fuglevand or leaky-integrate-and-fire) – generates MU spike trains from normalised neural drive, honouring the size principle and onion-skin firing behaviour.

5. **Signal synthesiser** – convolves MUAPs with the spike trains and sums across units/electrodes to yield HD-EMG signals.

6. **Utility toolbox** – helpers for movement definition, parameter conversion, spike-train visualisation and automatic data export.

**Technical details of EMG generation**  For benchmark generation, we restricted the MSK model to two degrees of freedom: wrist flexion–extension and radial–ulnar deviation. Each simulated gesture—whether static or dynamic—was confined to a single degree of freedom. MUAP templates corresponding to the target joint positions were generated using the *ab initio* mode of BioMime. Each synthetic recording contains EMG signals from a single superficial forearm muscle during the execution of a gesture.

We simulated 10 synthetic subjects, each defined by a static MU pool configuration across all muscles. For every subject, a fixed number of MUs was randomly sampled, along with individualised BioMime parameters [11] and a subject-specific fibre density. These parameters remained constant across all recordings for a given subject. Each subject is uniquely identified by its corresponding random seed.

To generate the complete benchmark set, we systematically varied the following parameters:

- **Movement DoF:** Flexion–extension or radial–ulnar deviation.
- **Muscle:** One of the 8 superficial forearm muscles listed below.
- **Effort profile:** One of four time-varying effort profiles: trapezoidal, sinusoidal, triangular, or ballistic.
- **Noise level:** Signal-to-noise ratio (SNR) sampled between 10 and 30 dB.
- **Electrode selection (NCols):** Number of electrode columns selected from the array. 32 columns cover the full ring; 5 or 10 columns provide a localized array around the target muscle.

## B.2   Latin Hypercube Sampling Parameter Configuration

The NeuroMotion configuration generation employs a two-level Latin Hypercube Sampling approach with common parameters applied across all movement profiles and profile-specific parameters varying by movement type.

### B.2.1   Parameter Definitions

**Common Parameters**  Six parameters are sampled uniformly across all movement profiles (see Supplementary Table 1).



Supplementary Table 1: Common LHS Parameters

| Parameter | Range | Units | Description |
|-----------|-------|-------|-------------|
| SubjectSeed | (0, 10) | – | Subject identification index |
| TargetMuscle | (0, 7) | – | Muscle selection index |
| MovementDOF | (0, 2) | – | Movement degrees of freedom |
| NCols | (0, 3) | – | Electrode column configuration |
| NoiseSeed | (1, 1000) | – | Noise generation seed |
| NoiseLeveldb | (10, 30) | SNR | Recording noise level |



**Movement-Specific Parameters**  Each movement profile requires distinct parameter sets, detailed in Supplementary Table 2.

Supplementary Table 2: Movement Profile-Specific Parameters

| Profile | Parameter | Range | Units | Notes |
|---|---|---|---|---|
| TRP Iso | EffortLevel | (5, 80) | % MVC | Muscle activation level |
| | RestDuration | (1, 3) | s | Pre/post rest period |
| | RampDuration | (5, 10) | s | Rise/fall time |
| | HoldDuration | (15, 30) | s | Sustained contraction |
| Sinuisoid Iso | EffortLevel | (15, 80) | % MVC | Base activation level |
| | RestDuration | (1, 3) | s | Pre/post rest period |
| | HoldDuration | (15, 30) | s | Total modulation time |
| | RampDuration | (5, 10) | s | Transition duration |
| | SinFrequency | (0.025, 0.5) | Hz | Modulation frequency |
| | SinAmplitude | (5, 15) | % MVC | Oscillation amplitude |
| Tri Iso | EffortLevel | (5, 80) | % MVC | Peak activation level |
| | RestDuration | (1, 3) | s | Pre/post rest period |
| | RampDuration | (1, 20) | s | Total triangle duration |
| Blt Iso | EffortLevel | (40, 100) | % MVC | Peak activation level |
| | RestDuration | (1, 3) | s | Inter-burst interval |
| | NRepetitions | (1, 30) | – | Number of bursts |
| Sinusoid Dyn | EffortLevel | (5, 80) | % MVC | Constant effort level |
| | TargetAnglePercentage | (0.5, 1) | % max | Movement amplitude |
| | TargetAngleDirection | (0, 1) | – | Direction index |
| | SinFrequency | (0.025, 0.5) | Hz | Angular oscillation rate |
| | SinAmplitude | (0.1, 0.5) | % max | Angle modulation depth |
| | HoldDuration | (10, 30) | s | Total movement time |
| Tri Dyn | EffortLevel | (5, 80) | % MVC | Constant effort level |
| | TargetAnglePercentage | (0.3, 1) | % max | Movement amplitude |
| | TargetAngleDirection | (0, 1) | – | Direction index |
| | RampDuration | (1, 6) | s | Movement duration |
| | NRepetitions | (1, 5) | – | Cycle repetitions |

### B.2.2  Sampling Strategy

**Profile Weighting**  Movement profiles are sampled with predetermined probabilities reflecting biomechanical prevalence:

$$P(\text{profile}) = P(\text{type}) \times P(\text{pattern} \mid \text{type}) \tag{S.1}$$

where $P(\text{Isometric}) = 0.65$ and $P(\text{Dynamic}) = 0.35$, yielding:

$$P(\text{Trapezoid\_Iso}) = 0.325 \qquad\qquad P(\text{Triangular\_Dyn}) = 0.175 \tag{S.2}$$
$$P(\text{Sinusoid\_Iso}) = 0.1625 \qquad\qquad P(\text{Sinusoid\_Dyn}) = 0.175 \tag{S.3}$$
$$P(\text{Triangular\_Iso}) = 0.08125 \tag{S.4}$$
$$P(\text{Ballistic\_Iso}) = 0.08125 \tag{S.5}$$

**System Constants**  The sampling framework incorporates fixed anatomical and technical constraints:

- **Muscles**: ECRB, ECRL, ECU, EDI, PL, FCU, FDSI
- **DOF ranges (degrees)**: Flexion-extension $(-65, 65)$, radial-ulnar $(-10, 25)$

- **Electrode arrays**: 5, 10, or 32 columns (32 chosen 60% of cases)

- **MU counts**: Normal distributions per muscle ($\mu \in [158, 422]$, $\sigma = 0.15\mu$)

## B.3 Hybrid Tibialis Anterior

Alongside the fully–synthetic data produced with NeuroMotion—where both the MUAPs and their driving spike trains are generated in silico—we curated a hybrid portion of the benchmark that grafts experimentally derived MUAP waveforms onto simulated neural drive. Using the output of HD-EMG decomposition together with spike-triggered averaging of 25 ms window, **1 070** unique MUs were extracted from ten able-bodied participants; the resulting MUAP templates were further processed so that they have compact support (using a Tukey window) and are zero mean. Thus, the MUAP templates are approximations of the true waveforms [1]. These templates were pooled, and we sampled between **70 and 150** units, for each synthetic subject, according to the exponential recruitment-threshold rule that mirrors Henneman's size principle [8]. For this dataset, we only generated isometric contractions. The NeuroMotion motor neuron pool recruitment model was used to simulate the neural activity.

We refer to this dataset as **Hybrid TA**. It comprises **five** synthetic subjects, each performing **20** contraction trials identical to those in the fully synthetic set.

## B.4 Motor Neuron Recruitment Model:

The default NeuroMotion settings were used to simulate the spike trains through the motor neuron recruitment model. The parameters are listed in the Supplementary Table 3:

Supplementary Table 3: Default Parameters for Motor Neuron Pool Class

| Parameter | Value | Description | Units |
|---|---|---|---|
| rr | 50 | Recruitment range: ratio of largest to smallest MU recruitment threshold | – |
| rm | 0.75 | Recruitment maximum: excitation level when all MUs are active | – |
| rp | 100 | Force fold: ratio of largest to smallest MU force capacity | – |
| pfr1 | 40 | Peak firing rate of the first (smallest) MU | Hz |
| pfrd | 10 | Peak firing rate difference between first and last MU | Hz |
| mfr1 | 10 | Minimum firing rate of the first (smallest) MU | Hz |
| mfrd | 5 | Minimum firing rate difference between first and last MU | Hz |
| gain | 30 | Excitatory drive-firing rate relationship (equivalent to 0.3 Hz per % MVC) | Hz/drive |
| c_ipi | 0.1 | Coefficient of variation for inter-pulse interval standard deviation | – |
| frs1 | 50 | Slope of drive-firing rate relationship for the first MU | Hz/drive |
| frsd | 20 | Difference in drive-firing rate slope between first and last MU | Hz/drive |

# C Experimental datasets

## C.1 Subjects

The three curated experimental datasets in MUniverse are Caillet et al. [4], Avrillon et al. [2], and Grison et al. [7], which include healthy subjects with no history of lower limb injury or pain during the months preceding the experiments. These datasets include ankle dorsiflexion experiments, where the Avrillon et al. dataset also includes a knee extension experiment. Caillet et al. dataset includes six healthy male subjects (26 ± 4 years; 174 ± 7 cm; 66 ± 15 kg), whereas the Avrillon et al. dataset includes eight healthy subjects for the ankle dorsiflexion experiment (27 ± 3 years) and eight for the knee extension experiment (27 ± 10 years). Grison et al. dataset includes one healthy male subject (39 years).

The respective Research Ethics Committees approved each study: Caillet et al. (Imperial College London, reference number 18IC4685), Avrillon et al. (Imperial College London, reference number 18IC4685; Comité de Protection des Personnes Ouest, reference number 23.00453.000166), and Grison et al. (Imperial College London, reference number 19IC5640). The studies followed the Declaration of Helsinki, and the subjects gave informed written consent.

## C.2 Experimental setups

A series of isometric ankle dorsiflexions (or knee extensions) was performed at given percentages of the MVC while recording EMG signals from the tibialis anterior (or vastus lateralis) muscle.

For the ankle dorsiflexions, participants sat with the hips flexed at 30 [4] or 45 [2] degrees, with 0 degrees the neutral hip position and knees fully extended. The right foot was fixed onto an ankle dynamometer (NEG1, OT Bioelettronica, Turin, Italy) positioned at 30 degrees in the plantarflexion direction, with 0 degrees being the foot perpendicular to the shank. The thigh and the foot were fixed with Velcro straps.

For the knee extensions, participants sat with the hips and knees flexed at 85 degrees [2], with 0 degrees being the neutral hip position and knees fully extended. The torso and thighs were fixed with Velcro straps, and the tibia was positioned against a rigid resistance connected to force sensors (Metitur, Jyväskylä, Finland). The force signals were recorded using the same acquisition system as the EMG recordings (see below).

A warm-up of sub-maximal isometric contractions was performed before each subject performed at least two MVCs. The maximal torque was used throughout the rest of the experimental session with a series of sub-maximal isometric contractions following a pattern of trapezoidal target paths, with 5% MVC/s ramps, displayed on a computer screen in real time, with the force level and target path visualised. In Caillet et al., trapezoidal paths of 30 and 50% MVC were used (with 20 and 15 s plateaus). In Avrillon et al., 10 to 80% MVC with 10% MVC increments were used (with 20, 15, and 10 s plateaus at 10-40%, 50-60%, and 70-80% MVC, respectively). Finally, in Grison et al., 10, 15, 20, 25, 30, 35, 40, 50, 60, and 70% MVC were used (with 20, 15, and 10 s plateaus at 10-30%, 35-40%, and 50-70% MVC, respectively). The order of force levels was randomly assigned, and there was a rest period between the tasks.

## C.3 EMG recordings

In the Caillet et al. and Avrillon et al. datasets, surface EMG signals were recorded from the TA (or the VL) muscle using four 64-channel grids (GR04MM1305 / GR08MM1305 for the TA / VL muscle, 13x5 gold-coated electrode configuration with a 4 / 8 mm inter-electrode distance; OT Bioelettronica, Italy). The Grison et al. dataset used two 64-channel grids with a 4 mm inter-electrode distance. The grids were placed over the muscle bellies identified by manual palpation. Before placing the electrodes, the skin was shaved and cleaned with 70% ethyl alcohol (Caillet et al.) or an abrasive pad and water (Avrillon et al.). A double-adhesive foam was attached to the electrode and the skin with conductive cream filling the adhesive layers' cavities, along with tape and elastic bands to secure the electro-to-skin contact. Bands damped with water were placed around the ankle as a patient reference.

Grison et al. dataset also includes implanted three 40-channel high-density intramuscular EMG arrays [15], oriented longitudinally and placed approximately 3 cm apart. The platinum electrodes

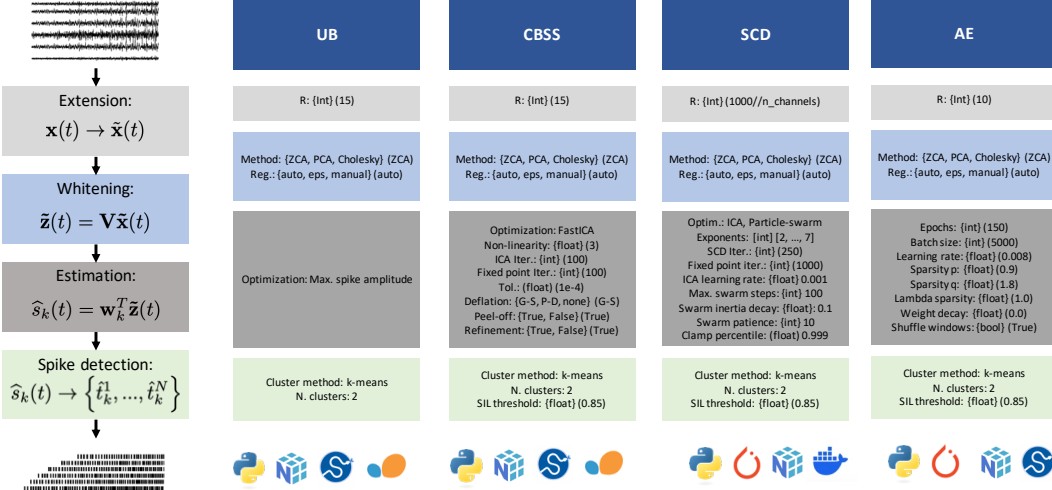

Supplementary Figure 2: **Overview of tested algorithms.** Each box provides a summary of the most important hyperparameters corresponding to the algorithm steps shown on the left. Abbreviations: G-S: Gram-Schmidt, P-D: Projection-deflation

are arranged on two sides of a filament (2x20 electrodes spaced 1 mm apart with 0.5 mm offset between sides). The concurrent surface and intramuscular EMG signal recording enabled two-source validation [5], where the intramuscular EMG signals were decomposed using SCD [6].

The EMG signals were recorded in monopolar derivation with a sampling frequency of 2048 Hz (surface EMG) or 10240 Hz (intramuscular EMG), amplified (150x), band-pass filtered (10–500 Hz for surface, 10–4400 Hz for intramuscular), and digitised using a 400-channel acquisition system with a 16-bit resolution (EMG-Quattrocento; OT Bioelettronica, Italy).

# D    Algorithm implementation

The MUniverse decomposition module currently provides three natively implemented algorithms (UB, CBSS and AE), together with a library of decomposition-specific general-purpose functionalities that facilitate the implementation of new algorithms. The native MUniverse decomposition functionalities are built on top of essential Python packages for scientific computing (NumPy, SciPy, scikit-learn, torch) and enable reasonably efficient computations while remaining accessible for a wide audience of users and interoperability across various computational environments. Further, external algorithms implemented in arbitrary programming languages/environments can be integrated, and here, e.g., a containerized version of the SCD algorithm is provided. An overview of the algorithms tested in the proposed manuscript is provided in Supplementary Figure 2.

## D.1    Upper-bound decomposition

The upper-bound algorithm estimates MU spike trains, making use of known MU impulse responses (see Section 2.2), and thus is only available for simulated datasets. Hence, the relevant hyperparameters only include the selected extension factor $R$, the whitening method (options: ZCA-whitening, PCA-whitening, or Cholesky-whitening), together with the regularization method of the whitening method (options: smallest-half of the eigenvalues, machine precision, arbitrary user-defined non-negative value). The projection vector is calculated by maximizing the expected spike amplitude given the extended and whitened MUAPs. Thus, for the upper-bound algorithm, the number of estimated sources is always equal to the number of MUAPs that are input to the algorithm. Although a source estimate always exists, the reconstruction of the activity of a full MU pool is typically not possible, as particularly small sources are covered by noise [10].

## D.2 CBSS

The implemented CBSS algorithm closely follows a standard ICA pipeline, which was first applied to MU decomposition in [16]. For experimental signals, one typically tries to reject parts of the noise through filtering. Thus, a bandpass (specifiable bandwidth and order) and a notch filter (adjustable power line frequency, number of harmonics, and order) are directly integrated into the decomposition pipeline. The following steps, i.e., extension and whitening, are shared with the upper-bound algorithm. For blindly optimizing the separation vector $\mathbf{w}_k$ given an objective function with user-tunable non-linearity (see Equation (4), the fastICA fixed-point algorithm is used [9]. Besides the parameter $a$ controlling the degree of non-linearity (note $a = 3$ is equivalent to using kurtosis as an optimization goal), the fixed-point algorithm has several hyper parameters: the number of sources to be extracted, the initialization strategy (random weights or activity index), convergence tolerance, and maximum number of iterations of the fixed-point algorithm as well as sub-space projection to avoid repeated convergence to already identified sources (Gram-Schmidt orthogonalization, projection deflation or none). Further, the estimated sources can be refined by recalculating the separation vector as the mean whitened signal at the estimated spike times, and high-quality sources (specifiable through a silhouette score threshold) can be peeled off to facilitate the identification of new sources. Lastly, the algorithm automatically classifies the identified sources into physiological or bad sources, whereby only sources with a user-specified minimum number of spikes and a minimum silhouette score are accepted.

## D.3 SCD

MUniverse provides a simple and clean API to the Swarm-Contrastive Decomposition algorithm [6, 7] (SCD), which extends classic ICA-based approaches in two key ways: (1) it adaptively selects the contrast function through an outer particle-swarm-optimisation loop that maximises source independence given the statistics of the current residual signal, and (2) it employs a peel-off strategy that removes each accepted MU source before the next optimisation round, preventing repeated convergence on the same dominant unit and exposing weaker ones. These extensions introduce additional hyperparameters, notably those governing the swarm (size, inertia schedule, cognitive/social coefficients, early-stopping patience) and the peel-off acceptance criteria (minimum-spike count and silhouette threshold). In this work, we keep all the particle-swarm optimization-related hyperparameters fixed, based on expert knowledge.

## D.4 AE

The implemented AE algorithm follows the fully unsupervised deep-learning approach proposed by [14], which formulates ICA as an autoencoder problem. The encoder is constrained to an orthogonal rotation that maps extended and whitened EMG observations into latent activations, each corresponding to an individual motor unit spike train. The decoder consists of a linear layer followed by a $\tanh$-shrink nonlinearity, reconstructing the whitened observations from the latent representation. The preprocessing stage applies a bandpass filter (20–500 Hz, 2nd order) and notch filtering at 50 Hz and its harmonics, matching the preprocessing used in CBSS. Training minimizes a combination of reconstruction loss and a sparsity penalty on the latents, parameterized by $(p = 0.9, q = 1.8, \lambda = 1.0)$. The network is trained for 150 epochs with a batch size of 5000 and a learning rate of 0.008 on GPU, using shuffled extended windows for stable convergence, all other hyperparameters are seeded from the original paper. Post-processing of the identified sources also follows the methods applied in CBSS.

# E  Notation and Acronyms

| Acronym | Full Form | Context/Description |
| --- | --- | --- |
| **BIDS** | Brain Imaging Data Structure | Community standard for organizing neuroimaging data and metadata |
| **Blt Iso** | Ballistic Isometric | Brief, high-intensity muscle contractions |
| **BSS** | Blind Source Separation | Signal processing technique for separating mixed signals |
| **CBSS** | Convolutive Blind Source Separation | Algorithm for EMG decomposition based on FastICA |
| **CC-BY 4.0** | Creative Commons Attribution 4.0 | Open license type |
| **dB** | Decibel | Signal amplitude unit |
| **DoF** | Degrees of Freedom | Movement parameters (e.g., wrist flexion-extension) |
| **ECRB** | Extensor Carpi Radialis Brevis | Forearm muscle |
| **ECRL** | Extensor Carpi Radialis Longus | Forearm muscle |
| **ECU** | Extensor Carpi Ulnaris | Forearm muscle |
| **EDF** | European Data Format | Standardized file format for biological signals |
| **EDI** | Extensor Digitorum Indicis | Forearm muscle |
| **EEG** | Electroencephalography | Brain electrical activity recording |
| **EMG** | Electromyography | Muscle electrical activity recording |
| **FAIR** | Findable, Accessible, Interoperable, Reusable | Data management principles |
| **FastICA** | Fast Independent Component Analysis | Algorithm for blind source separation |
| **FCU** | Flexor Carpi Ulnaris | Forearm muscle |
| **FDSI** | Flexor Digitorum Superficialis Indicis | Forearm muscle |
| **FEM** | Finite Element Method | Numerical method for solving differential equations |
| **FN** | False Negative | Evaluation metric for spike detection |
| **FP** | False Positive | Evaluation metric for spike detection |
| **FVE** | Fraction of Variance Explained | Performance metric for decomposition quality |
| **gCKC** | Gradient Convolution Kernel Compensation | EMG decomposition algorithm |
| **G-S** | Gram-Schmidt | Orthogonalization method |
| **HD-EMG** | High-Density Electromyography | Multi-channel EMG recording technique |
| **Hz** | Hertz | Frequency unit |
| **ICA** | Independent Component Analysis | Statistical method for signal separation |
| **IRB** | Institutional Review Board | Ethics committee for human subjects research |
| **JSON** | JavaScript Object Notation | Data interchange format |
| **JSON-LD** | JSON Linked Data | Structured data format |
| **LHS** | Latin Hypercube Sampling | Statistical sampling technique |
| **LLM** | Large Language Model | AI language processing system |
| **MEG** | Magnetoencephalography | Brain magnetic field recording |
| **MNE** | MNE-Python | Software package for neurophysiological data analysis |
| **MRI** | Magnetic Resonance Imaging | Medical imaging technique |
| **ms** | Millisecond | Time unit |
| **MSK** | Musculoskeletal | Related to muscles and skeleton |
| **MU** | Motor Unit | Smallest voluntarily contractible unit (motor neuron + muscle fibers) |

**Table 4 continued from previous page**

| Acronym | Full Form | Context/Description |
|---|---|---|
| MUAP | Motor Unit Action Potential | Electrical signal from motor unit activation |
| MVC | Maximum Voluntary Contraction | Peak muscle force capability |
| NA | Not Applicable | Response option in paper checklist |
| PCA | Principal Component Analysis | Dimensionality reduction technique |
| P-D | Projection-Deflation | Source separation technique |
| PD III | Precision Decomposition III | Template matching algorithm |
| PD-IGAT | PD with Integrated Grouping and Template matching | Extended template matching algorithm |
| PD-IPUS | PD with Integrated Pulse and Template matching | Template matching algorithm variant |
| PL | Palmaris Longus | Forearm muscle |
| PNR | Pulse-to-Noise Ratio | Signal quality metric |
| REPL | Read-Eval-Print Loop | Interactive programming environment |
| RoA | Rate of Agreement | Performance metric for spike matching |
| SCD | Swarm-Contrastive Decomposition | EMG decomposition algorithm using particle swarm optimization |
| sEMG | Surface Electromyography | Non-invasive EMG recording from skin surface |
| SIL | Silhouette Score | Clustering quality metric |
| Sinusoid Dyn | Sinusoidal Dynamic | Sinusoidal effort modulation, dynamic movement |
| Sinusoid Iso | Sinusoidal Isometric | Sinusoidal effort modulation, isometric |
| SNR | Signal-to-Noise Ratio | Measure of signal quality |
| TA | Tibialis Anterior | Lower leg muscle |
| TCL | Time-Contrastive Learning | Deep learning approach for source separation |
| TP | True Positive | Evaluation metric for spike detection |
| Tri Dyn | Triangular Dynamic | Triangular-shaped effort profile, dynamic movement |
| Tri Iso | Triangular Isometric | Triangular-shaped effort profile, isometric |
| TRP | Trapezoidal | Type of contraction profile |
| TSV | Tab-Separated Values | File format for tabular data |
| ZCA | Zero-phase Component Analysis | Whitening transformation method |
| $\mu$ | Mean | Statistical parameter |
| $\sigma$ | Standard Deviation | Statistical parameter |