# OpenReview forum: "MUniverse: A Simulation and Benchmarking Suite for Motor Unit Decomposition"
_NeurIPS.cc/2025/Datasets_and_Benchmarks_Track — NeurIPS 2025 Datasets and Benchmarks Track poster_

### Official Review · Reviewer_TsqS · 2025-06-29

**Rating:** 5
**Confidence:** 3

**Summary:**

This work introduces MUniverse, a single repository for benchmarking motor unit decomposition methods. Its scope covers the curation of several EMG dataset sources, basic methods for MU decomposition, and comprehensive evaluation metrics. There is also notable effort to integrate the NeuroMotion simulator as a scalable data source in this framework.

**Additional Feedback:**

I have small questions about the exploratory experimental results.

Section 6: A useful outcome of a benchmark is to identify relative strengths and weaknesses of different methods. Currently, Table 2 shows an edge to SCD but Table 3 shows an edge to CBSS (at least by F1). Could there be some discussion of the result in 6.3 given it contrasts with 6.2? If precision/recall/F1 is dependent on the number of MUs recovered, is it important to be able to compare across methods while matching MUs?

Table 2/3. The 10/50/90 percentiles show very large variability in pipeline performance. Is this variance across datasets or different runs of an algorithm? I would assume the latter, but L347 says there is minimal and expert hyperparameter selection, so I am not clear how this selection can be so variable across random seeds?

More generally, is this package intended to recommend a single method for external users to take and use off the shelf? While having many metrics is useful for domain experts, it wasn't very easy to discriminate even two methods in overall performance even in the small evaluation here. If different methods are expected to excel at different settings, then will users be expected to judge which datasets are most relevant for them?

**Dataset Code Accessibility:**

Yes

**Dataset Code Comments:**

I have browsed the code package and verified the stated datasets are available in the provided data repository. There is a metadata trail to the versions of the previous data release before their conversion for this work.

**Ethical Comments:**

No. All experimental data was collected in previously published work, this particular work simply reformats them for community use. Notably, there is a callout to ensure method performance across populations but I do not see any particular translation of this principle in the work.

**Ethical Considerations:**

No, there are no or only very minor ethics concerns

**Final Justification:**

My two critiques were in clarity and significance. I believe the work will become more significant with the stated future expansion (achieving both breadth and depth), and the rebuttal response indicates the manuscript will also be revised to more clearly describe the shape of MUniverse's contributions. I am happy with this work as a forward looking whitepaper for this subfield.

**Limitations Weaknesses:**

Quality and originality discussed above.

**Clarity**: My clarity concern is that it is difficult to parse what work has been done beyond existing resources. For example, while I infer that MUniverse is designed for MU data science and NeuroMotion is designed for simulation, it'd be useful to understand how the user-friendly interface defined by MUniverse differs from the lower-level one defined by NeuroMotion. I also suggest that given that the simulation component of MUniverse is highlighted prominently in summary text, it would be valuable to show some simulation results in the main text, and e.g. move 5.1 to appendix. There are also a few smaller writing nits that I was confused by:

- Section 2: Given the focus of the work as a benchmark/dataset contribution, the related works section seems to overemphasize prior BSS methods relative to prior ecosystem work (which MUniverse advances). This is especially important given how many areas MUniverse aims to address. It would be useful context to know the scope or adoption of prior open-source toolboxes, simulators, or datasets, even if they are limited. A table might be one way to accomplish this. I'd be happy with such a table in the appendix or the webpage.
- Fig 2: Having a caption describe 3 modules in a figure with four boxes is confusing.
- l95-96: Are this two or three assumptions? I got lost by the phrasing. Since these theoretical assumptions are raised, I am also curious whether the assumptions are met in practice given statistical independence of sources seems quite unlikely.
- l162: Mentioning that SCD is containerized seems extraneous as it is still unclear what SCD is other than an external package (a callout to the later definition would be helpful).
- Section 4.2: For non-experts (like me), it would be useful to contextualize why the specific 10K/1K simulation sizes were chosen (these numbers don't seem particularly large or burdensome), and in what sense they are exhaustive. In a similar vein, there are a number of hyperparameters in the paper that could use some minimal rationale. For example, why is 5-80% MVC simulated as opposed to 0-100%? If those choices are typically made for experimental convenience, it's not clear they should also be respected in simulation. On L309, a 30% overlap in common spikes seems low for ground truth assignment. Is this considered a conservative threshold for commercial or experimental hardware?

I expect to improve my clarity rating after these comments are addressed in the review process.


**Significance**: The significance of this work will depend on community adoption. The current vision for how MUniverse is meant to be used by the community is only somewhat compelling. This is because the work has large breadth across the MU stack, but seems to have traded for a shallow depth (3 datasets and 2 methods seed the repo). While there is a stated invitation for the community to engage, this initialization may not already entice users focused on any one part of the stack. New method-makers might not learn the framework to only test their method on a few new datasets, and new dataset contributors might not see the marginal value of formatting their dataset for compatibility with MUniverse, given these developers likely already have internal tooling of similar scale. This small seeding combines with the currently limited technical scope, which I understand less. To mitigate this weakness, I suggest adding some context and timescale to the future work section that describes which major features are planned to reach a version 1.0 (if this is not already reached) that the community is expected to sustain and adopt.

**Strengths Contributions:**

**Quality**: The work covers many aspects of the MU decomposition problem and each component appears to be treated with care. The dataset release and code package are well organized (though 3 of the 4 notebooks did not load on the code provided).

**Originality**: A single cross-cutting resource for covering the MU decomposition problem is an interesting proposal. It is unclear to me that these components will cohere well in the long term. For example, currently it seems like it took some effort to curate and convert pre-existing EMG datasets into a uniform interface for MUniverse, but if consensus on dataset release becomes more standardized, it's not clear there will remain a distinction between a "raw" dataset and a "curated" dataset, implying MUniverse's role will be primarily in evaluation and benchmarking. Similarly, simulation and physical datasets proceed in robotics and neuroscience along separate tracks, so I'm not clear if their merging here is particularly valuable. Nonetheless, I appreciate the effort to build consensus in any human ephys domain.

Significance and clarity discussed below.

---

> ### Author Rebuttal · Authors · 2025-07-30
>
> We sincerely thank the reviewer for their thorough and constructive review. Your comments provided valuable guidance on clarifying our contributions, improving presentation, and outlining our roadmap. We emphasize that MUniverse is the first complete benchmarking ecosystem for motor unit decomposition unifying simulation, curated datasets, baseline algorithms, and standardized evaluation metrics. While the current version does seed the benchmark with a smaller but diverse set of datasets and methods, we strive to expand both breadth and depth, ensuring long-term community impact. Below, we address each point in detail.
>
> > *Quality* ... The dataset release and code package are well organized (though 3 of the 4 notebooks did not load on the code provided).
>
> We apologize for this issue. The three notebooks were inadvertently uploaded in the final repository version. We have corrected this in our current branch and will ensure that functional, well-documented tutorials are released by the camera-ready version.
>
> > *Originality:* A single cross-cutting resource for covering the MU decomposition problem is an interesting proposal. It is unclear to me that these components will cohere well in the long term. For example, ...
>
> We appreciate this thoughtful observation.  (a) On EMG data standards and curation: we note that standardization and curation are two complementary strengths of MUniverse. By adopting the emerging HD-EMG BIDS standard, we ensure that datasets remain easy to maintain and integrate. The curation component brings together diverse datasets spanning muscles, contractions, recording setups, and synthetic conditions. We believe this diversity will remain a valuable strength, even as standards evolve, as MUniverse lowers the barrier for method developers to access heterogeneous datasets through a single interface and readily compare the performance against well-established algorithms.
>
> (b) On merging simulated and experimental recordings: We view this integration as a major advantage of MUniverse. Historically, algorithm evaluations have been fragmented, often relying on private datasets without ground truth. MUniverse enables a more rigorous and transparent workflow by combining simulated data (with known ground truth) and experimental data (for real-world applicability). As each of these approaches has unique strengths, this dual focus provides a more holistic benchmark for decomposition algorithms, similar to efforts in intracortical spike-sorting literature (Magland et. al. 2020).
>
> > *Clarity:* My clarity concern is that it is difficult to parse what work has been done beyond existing resources.
>
> We thank the reviewer for this suggestion and will move selected simulation results from the supplementary material into the main text, utilizing the additional space available in the camera-ready version.
>
> Regarding the distinction from NeuroMotion: while NeuroMotion provides low-level control of EMG simulation (e.g., specifying detailed anatomy and motor unit properties), MUniverse builds a higher-level, user-friendly interface. Users can easily replicate common experimental setups, define force and joint angle trajectories, and customize motor unit properties via a single configuration file. Moreover, it enables the use of MUAP waveforms from other simulators or from experiments.
>
> > Given the focus of the work as a benchmark/dataset contribution, the related works section seems to overemphasize prior BSS methods ...
>
> We agree and will add a concise table in the supplementary comparing prior open-source tools, datasets, and simulators, highlighting how MUniverse integrates these components into the first end-to-end benchmark for motor unit decomposition. Specifically, we will note that open-source tooling for EMG exists as follows: (1) datasets: individual datasets shared publicly by authors (typically in a recording system-dependent output format), but no standardized datasets available until MUniverse, (2) open source simulators: NeuroMotion, Maier et. al. (2024), Costa-Garcia et. al. (2025), some community adoption but not trivial to use; MUniverse provides easy-to-simulate config-based routines and tutorials, (3) algorithms: mentioned in methods, will be repeated.
>
> > Fig 2: Having a caption describe 3 modules in a figure with four boxes is confusing.
>
> Thanks for highlighting that issue. We will revise the caption and harmonize terminology between Section 3.2, the code modules, and the figure layout for consistency.
>
> > l95-96: Are this two or three assumptions?...
>
> There are three assumptions in the ICA model: (i) the sources are linearly mixed, (ii) no more than one source is Gaussian, and (iii) the joint distribution of sources factorizes. While strict independence is not satisfied in EMG, motor unit spike trains have weak practical dependence (i.e., due to sparsity measures such as mutual information being low, although spike trains have latent dependencies) and are super-Gaussian, which allows ICA-based methods to perform effectively. In practice, ICA loss functions optimize for non-Gaussianity, which reliably separates motor units that have high amplitude, see Klotz and Rohlen (2025).
>
> > l162: Mentioning that SCD is containerized seems extraneous ...
>
> Thanks for pointing out that issue. We will revise Section 2.1 to introduce SCD more clearly at first mention and move implementation details, such as containerization, to the implementation section.
>
> > Section 4.2: For non-experts (like me), it would be useful to contextualize why the specific 10K/1K simulation sizes were chosen...
>
> Thanks for giving us an opportunity to clarify these aspects. The reasoning behind most of the said choices boils down to resource constraints. (1) The 10K dataset was chosen to comprehensively cover a range of recording conditions and configurations of interest, while maintaining manageable storage (hundreds of GBs). The 1K dataset is a smaller subset, enabling reproducible algorithm runs within approximately 1000 CPU hours. (2) For the simulated test datasets, we limited MVC to 5–80\% to (i.) better reflect experimental conditions, and (ii.) corresponding to the regime where the model is considered valid (e.g., the simulation does not resolve fatigue-related changes in MUAP waveforms that become increasingly important with increasing MVC level). Nevertheless, the training dataset is more explorative and spans a broader range. (3) The 30\% overlap threshold follows established literature and is considered a conservative, standard rule of thumb for matching decomposed units to ground-truth (e.g., Negro et. al 2016).
>
> > Significance: The significance of this work will depend on community adoption...
>
> We agree that community adoption is central to MUniverse’s significance. While we already include 5 test datasets (plus 1 large train dataset) and two widely-used algorithms (CBSS and SCD), expanding this foundation is a top priority.
>     Our roadmap for version 1.0 (next 12 months) includes:
>     (i) expanding the dataset library to 10–15 datasets spanning more muscles, tasks, and validated ground truth,
>     (ii) adding other algorithms (e.g., deep learning-based baselines) alongside CBSS and SCD
>     (iii) improving tutorials, notebooks, and pipelines for ease of adoption, and
>     (iv) launching a dynamic leaderboard that guides algorithm selection for different applications.
>
> > Section 6: A useful outcome of a benchmark is to identify relative strengths and weaknesses of different methods...
>
> We appreciate this observation. These results reflect differences in how metrics balance the number versus quality of recovered units. While it is possible to combine metrics into a single score, we chose to keep them separate to highlight complementary aspects of performance. We will add a short discussion of this contrast in Section 6.3.
>
> > Table 2/3. The 10/50/90 percentiles show very large variability in pipeline performance...
>
> The reported variability arises from performance differences across recordings within each dataset, not random seeds, as hyperparameters are fixed. It is well known that even for a single experimental protocol, the number of identified motor units can vary considerably between subjects. For example, due to subject-level factors such as the thickness of the subcutaneous fat tissue or the spatial distribution of the MU territories and sensing-related factors such as electrode-skin impedance or the position of the EMG-girds.  We will clarify this in the manuscript.
>
> > More generally, is this package intended to recommend a single method for external users to take and use off the shelf?...
>
> MUniverse is designed as a benchmarking platform rather than a prescriptive tool. Our focus is on standardized datasets (via BIDS) and reproducible metrics so that researchers can compare new methods against open baselines. Given the diversity of conditions, we expect that no single method will dominate all scenarios. In the future, our planned leaderboard will allow users to filter by muscle, condition, or task, helping both algorithm developers and practitioners identify suitable methods for their use cases.
>
> ---
>
> References
>
> Negro, F., et. al. (2016). Multi-channel intramuscular and surface EMG decomposition by convolutive blind source separation. Journal of neural engineering, 13(2), 026027.
> Magland, J., et. al. (2020). SpikeForest, reproducible web-facing ground-truth validation of automated neural spike sorters. Elife, 9, e55167.
> Maier, B., et. al. (2024). OpenDiHu: an efficient and scalable framework for biophysical simulations of the neuromuscular system. Journal of Computational Science, 79, 102291.
> Costa-Garcia, A., et. al. (2025). Tailoring neuromuscular dynamics: A modeling framework for realistic sEMG simulation. PLoS One, 20(6), e0319162.

---

> > ### Comment · Reviewer_TsqS · 2025-08-02
> >
> > Thank you for the thorough response. I believe my critiques have been well addressed. The future roadmap sounds like a substantial and exciting expansion.

---

> > > ### Author Response · Authors · 2025-08-07
> > >
> > > We thank the reviewer again for their thorough and constructive feedback, and share their enthusiasm for MUniverse's future roadmap.

---

### Official Review · Reviewer_1A1q · 2025-07-02

**Rating:** 5
**Confidence:** 2

**Summary:**

This paper presents MUniverse, a simulation and benchmark for motor unit decomposition. It contains a simulation stack and API to an EMG simulator, a library of synthetic, hybrid and experimental datasets, decomposition pipelines, and a benchmark with standardized tasks, evaluation metrics, and baseline performance comparison. The suite follows FAIR principles and standardized formats for data sharing.

**Dataset Code Accessibility:**

Yes

**Dataset Code Comments:**

I was able to access the shared datasets and associated code. They look well organized and sufficiently documented.

**Ethical Considerations:**

No, there are no or only very minor ethics concerns

**Final Justification:**

The paper provides a comprehensive suite of datasets and algorithms for motor unit decomposition. With the provisioned deep-learning based baselines, I think the paper will be further strengthened and appeal to a broader audience at NeurIPS. Therefore I recommend acceptance for the paper.

**Limitations Weaknesses:**

1. While appropriately acknowledged in the paper, the benchmark could benefit more from the inclusion of deep learning baselines. Approaches such as TCL and VAE mentioned in Section 2.2 could be good candidates, although even simple, trivial networks like a multilayer perceptron could serve as a baseline in this category.
2. Analyses on cross-session/cross-subject generalization were not mentioned in the paper, despite being an important issue. How do the presented algorithms perform across sessions/users?

**Strengths Contributions:**

1. The paper gives significant contribution to the research field studying motor unit decomposition from EMG signals, an important problem in neural signal processing.
2. The dataset and benchmark suite builds upon previous works, addressing the lack of open and diverse datasets in this research subfield.
3. The choices of simulator, datasets, evaluation metrics, and baselines are reasonable and well containerized.
4. The paper is well-written with sufficient technical details and follows a logical structure.

---

> ### Author Rebuttal · Authors · 2025-07-30
>
> We thank the reviewer for their thoughtful feedback and recognition of MUniverse’s contributions. We agree that adding more algorithms (e.g., deep learning baselines) would strengthen the benchmark and plan to address this in upcoming updates. Regarding cross-session and cross-subject generalization, we note that traditional decomposition algorithms are designed to fit individual recordings, but we see this as an exciting future direction, particularly with learning-based approaches.
>
> > While appropriately acknowledged in the paper, the benchmark could benefit more from the inclusion of deep learning baselines...
>
> We acknowledge that the paper would significantly benefit from a (deep) learning-based baseline. Given the requests of 2 of the 4 reviewers, we will include 2 learning-based algorithms in our benchmark by the camera-ready version. Despite not being a true blind- source separation method, we will include a deep-learning baseline that is trained on CBSS outputs (Clarke et. al. 2020), as it  represents the current state-of-the-art. In addition, we will implement one unsupervised deep-learning method, such as a vanilla implementation of TCL/ VAE, to extract "source-like" latents, followed by post-processing of the spike trains from current algorithms.
>
> > Analyses on cross-session/cross-subject generalization were not mentioned in the paper...
>
> This is an important point, but traditional decomposition pipelines are not designed to enable cross-session or cross-subject generalization. These algorithms fit data of a specific recording to the convolutive mixture model and therefore capture recording-specific activity rather than generalizing across recordings or datasets. We agree that this is an interesting challenge, and we are keen to explore how learning-based algorithms could enable such generalization in future work. However, for the current work, this is beyond the intended scope.
>
> ---
>
> References:
>
> Clarke, A. K., et. al. (2020). Deep learning for robust decomposition of high-density surface EMG signals. IEEE Transactions on Biomedical Engineering, 68(2), 526-534.

---

> > ### Comment · Reviewer_1A1q · 2025-08-07
> >
> > I thank the authors for their responses. I have no further concerns or questions. I think the addition of deep-learning baselines to the final version of the paper can add values to the paper. The authors can explain the caveats associated with these baselines for extra clarity as well.

---

> > > ### Author Response · Authors · 2025-08-07
> > >
> > > We thank the reviewer for their constructive feedback and look forward to analyzing how deep-learning baselines fare. We will also include a more thorough discussion of the caveats associated with learning-based decomposition pipelines in the camera-ready version.

---

### Official Review · Reviewer_FHTj · 2025-07-03

**Rating:** 4
**Confidence:** 4

**Summary:**

The MUniverse project presents a comprehensive simulation and benchmarking suite for motor unit (MU) decomposition from high-density EMG recordings. It addresses the lack of standardized datasets in the field by providing synthetic, hybrid, and experimental EMG datasets, with associated ground truth spike trains where available. The package includes multiple decomposition algorithms, baseline results, and evaluation metrics within a standardized and extensible software framework. The data is distributed following BIDS-EMG and Croissant metadata standards, and hosted publicly via Harvard Dataverse.

**Dataset Code Accessibility:**

Yes

**Dataset Code Comments:**

The dataset and code are publicly available through Harvard Dataverse and GitHub, respectively. The project includes a clear folder structure, setup instructions, and metadata formatted using BIDS-EMG and Croissant standards. A range of synthetic, hybrid, and experimental data is provided, along with preprocessing scripts, benchmark pipelines, and baseline results, enabling reproducibility and extensibility.

**Ethical Considerations:**

No, there are no or only very minor ethics concerns

**Final Justification:**

The submission presents a valuable benchmarking suite and dataset for motor unit decomposition, with clear potential impact and strong technical foundations. The inclusion of multiple data types, standard formats, and public code contributes positively to reproducibility and reuse.

However, based on the rebuttal and submission formatting, I have concerns regarding the level of care applied to this submission. The metadata in the original submission was incomplete, and while the authors acknowledged this and committed to correcting it, the rebuttal itself was casually formatted (e.g., markdown artifacts, unresolved placeholders, and inconsistent punctuation), which gives an impression of limited attention to detail.

The response also remained high-level, with future promises rather than concrete revisions. Given the high standards expected at this venue, I am adjusting my score to 4 to reflect these concerns, while still recognizing the value of the core contribution.

**Limitations Weaknesses:**

* Incomplete metadata in submission: Several sections (e.g., motivation, data collection) contain unresolved placeholder fields such as <authors list>, <institutions>, and <grant IDs>, which limit transparency regarding dataset authorship and funding.

* Experimental subject diversity: While the dataset includes experimental recordings, the total number of subjects is limited, and some tasks involve only one or a few individuals.

* Use scope limitations: The dataset is not validated for clinical decision-making or safety-related applications (e.g., neurostimulation), which should be clearly stated for downstream users.

**Strengths Contributions:**

* Well-organized codebase: Clear project structure and setup instructions, distributed as a Python package with modular components for generation, decomposition, and evaluation.

* Diverse and scalable datasets: Includes synthetic, hybrid, and experimental EMG signals across multiple tasks and muscles, with spike ground truth where available.

* Benchmarking framework: Provides multiple algorithms, unified metrics, and visualizations to enable performance comparison.

* Standardized formats: All data is organized under BIDS-EMG and Croissant metadata standards, supporting reproducibility and downstream integration.

* Public access and licensing: Code is MIT-licensed; data is released under CC-BY 4.0 via persistent DOIs on Harvard Dataverse.

* Used in publications: Experimental subsets have already been used in peer-reviewed studies (Caillet 2023, Avrillon 2024, Grison 2025), enhancing their credibility and reuse potential.

---

> ### Author Rebuttal · Authors · 2025-07-30
>
> We thank the reviewer for their positive evaluation and helpful feedback. We will complete all metadata fields, clarify dataset authorship and funding, and emphasize that the presented datasets are high-quality seed examples for a community-driven benchmark. We will also explicitly state that the dataset is not intended for clinical use and plan to expand dataset diversity in future releases. Please see below for a detailed response.
>
> > Incomplete metadata in submission...
>
> We acknowledge this issue and apologize for the omission. The datasets already include a comprehensive `dataset_description.json` file as required by the BIDS specification, which contains the correct metadata (authorship, funding, etc.). However, we missed copying these fields into the Dataverse metadata, resulting in the placeholders seen in the submission. We will ensure that this is corrected both in Dataverse and in the camera-ready version.
>
> > Experimental subject diversity...
>
> We acknowledge that one dataset includes only a single subject. This dataset combines high-density surface EMG with (invasive) high-density intramuscular EMG recordings, an approach that is valuable for obtaining ground-truth spike trains, thanks to the high selectivity of the latter. However, such experiments are highly challenging, costly, and are currently feasible only in a few specialized labs. Given the diversity of muscles, recording systems, and tasks, we believe that building a sufficiently comprehensive database requires a community-driven effort. Therefore, the datasets in MUniverse should be viewed as high-quality, representative examples of curated dataset classes, namely (i) simulations with known ground truth, (ii) paired invasive and surface EMG recordings, and (iii) pure surface EMG recordings with expert-annotated decompositions. In future releases, we intend to curate larger, more diverse datasets to accelerate method development and leverage the groundwork we have laid in standardising the datasets.}
>
> > Use scope limitations...
>
> We agree with this comment and will ensure that the manuscript explicitly states that the dataset is not validated for clinical or safety-critical use. While surface EMG-based motor unit identification methods hold promise for clinical applications, there are currently no standardized workflows using these methods.

---

> > ### Comment · Reviewer_FHTj · 2025-08-06
> >
> > Thank you for your response and for clarifying key aspects of the submission, including the metadata structure, dataset composition, and intended use scope. I appreciate the effort to address the points raised.
> >
> > That said, I would like to note some concerns regarding the quality of the rebuttal itself. The response contains visible formatting artifacts (e.g., markdown headers, curly braces, and incomplete section markers), and reads more like a high-level summary than a careful, targeted reply. While the core content of your work remains valuable, this presentation does affect the perception of rigor and care, especially for a resource-oriented submission intended for broad reuse.
> >
> > I encourage the authors to revise the camera-ready version with greater attention to detail, including resolving metadata issues across all hosting platforms and ensuring a more polished and professional presentation. This will improve the clarity, usability, and long-term impact of your contribution.

---

> > > ### Author Response · Authors · 2025-08-07
> > >
> > > We thank the reviewer for acknowledging the value of our work and for their encouragement and close attention to detail. We apologize for the formatting artifacts in our rebuttal. In preparation for the camera-ready version, we will resolve all metadata inconsistencies across hosting platforms and ensure that dataset descriptions, code documentation, and the paper are clear, complete, and consistent. As a team committed to rigor, openness, and long-term reuse, we take these concerns seriously and will actively work to ensure MUniverse becomes a high-quality, sustainable benchmark for the community.

---

### Official Review · Reviewer_zJjw · 2025-07-07

**Rating:** 4
**Confidence:** 4

**Summary:**

MUniverse introduces  open-source simulation and benchmarking suite designed to advance research in motor unit (MU) decomposition from electromyographic (EMG) signals. This platform has three core components: a simulation stack that utilizes the NeuroMotion simulator for generating realistic EMG data, a library of diverse datasets (including synthetic, hybrid, and experimental types), and a suite of decomposition pipelines, which encompass  algorithms such as CBSS and SCD. Additionally, MUniverse integrates a benchmarking framework equipped with established evaluation metrics and reporting functionalities. The suite is built with an emphasis on modularity, reproducibility, and adherence to FAIR and BIDS principles.

**Additional Feedback:**

1. Add some advance deeplearning methods in the benchmark.

2.On the Choice of CBSS Objective Function: Equation (4) presents a generalized contrast function for the FastICA-based implementation. What was the rationale for choosing this specific function over more traditional ones (e.g., skewness or kurtosis)? Did you perform any analysis to show its superiority for EMG decomposition?

**Dataset Code Accessibility:**

Yes

**Dataset Code Comments:**

The dataset is opensource and provide some code to reporduce the results.

**Ethical Considerations:**

No, there are no or only very minor ethics concerns

**Limitations Weaknesses:**

1. Evaluation of "Upper-Bound" Algorithm: The paper presents a "theoretical upper-bound estimate". However, the results section (Section 6) and the corresponding tables (Table 2 , Table 3 ) do not show any results for this upper-bound algorithm. Its performance is never actually reported, making its inclusion feel incomplete.

2. Absence of Deep Learning Baselines: The paper discusses deep learning approaches in the background section but doesn't include any in the actual benchmark.

3.Fixed Hyperparameters May Obscure True Algorithm Performance: The decision to use "expert-chosen hyperparameters"  for the evaluated algorithms (CBSS, SCD) is a significant limitation.

**Strengths Contributions:**

**Addresses a Critical Gap**: The paper directly tackles the absence of open benchmarks in EMG decomposition, a long-standing issue that has hindered progress in the field compared to related domains like spike sorting or EEG analysis. By providing MUniverse.

**Diverse Dataset Curation**: The curated library includes synthetic, hybrid synthetic-real, and experimental EMG data, covering a wide range of muscles, contraction types, recording configurations, and noise conditions.

**Well-written and Organized**: The paper is well-structured and is esay to understand.

---

> ### Author Rebuttal · Authors · 2025-07-30
>
> We thank the reviewer for the constructive feedback and for recognizing the value of MUniverse as a standardized, open-source benchmarking suite for motor unit decomposition. We appreciate the detailed comments on missing baselines, evaluation clarity, and hyperparameter selection. Below, we address each point and outline our planned improvements for the camera-ready version and future releases.
>
> > Evaluation of "Upper-Bound" Algorithm...
>
> Tables 2 and 3 report performance only on the experimental datasets, for which no ground-truth spike trains, as well as the corresponding motor unit impulse response waveforms, are available. Consequently, the upper-bound algorithm, which relies on known impulse responses, cannot be applied. Due to space constraints, we needed to make a decision on only presenting some exemplary results in the main text (i.e., where we selected decomposition outputs from the experimental datasets because the EMG community gives more weight to this datatype). Fortunately, the camera-ready version provides one additional page, and in line with comments from other reviewers, we will move the supplementary Table 1, which includes results of the upper-bound method for simulated data, into the main text.
>
> > Absence of Deep Learning Baselines:
>
> We acknowledge that the paper would significantly benefit from a (deep) learning-based baseline. However, we left this out for the following reasons: (1) previous deep learning based methods, as mentioned in the background section, are all trained on outputs of CBSS algorithms in a supervised fashion. These methods, therefore, are geared towards source separation of already identified units in new recordings of the exact same subject/ task/ recording setup. As such, there are no proposed methods to directly solve the (unsupervised) decomposition problem of converting EMG signals into spike trains. (2) Given MUniverse's focus on (a) standardizing dataset availability, (b) standardizing benchmarking metrics and routines, and (c) introducing a framework that is reproducible and easy to use, we have unfortunately left out implementing new algorithms by ourselves so far.
>
> However, given the requests of 2 of the 4 reviewers, we will include 2 learning-based algorithms in our benchmark by the camera-ready version. Despite not being a true blind- source separation method, we will include a deep-learning baseline that is trained on CBSS outputs (Clarke et. al. 2020), as it  represents the current state-of-the-art. In addition, we will implement one unsupervised deep-learning method, such as a vanilla implementation of TCL/ VAE, to extract "source-like" latents, followed by post-processing of the spike trains from current algorithms.
>
> > Fixed Hyperparameters May Obscure True Algorithm Performance...
>
> We agree that this is a limitation. This was a deliberate trade-off due to the time requirements of current decomposition algorithms. From Table 1, the median runtime for both CBSS and SCD on a single recording is around 600–800 seconds. Prior work indicates that hyperparameters may need tuning depending on the recording setup and task, often at the single-recording level. Even basic hyperparameter optimization routines (e.g., Bayesian optimization of a few key parameters) would require substantial computational resources beyond our capacity. We therefore opted to use a single set of expert-chosen hyperparameters. Although we believe that the selected hyperparameters are a good choice, there is no guarantee of optimality.
>
> We will therefore include a preliminary hyperparameter optimization analysis by perturbing a few of the most important hyperparameters per algorithm, on a subset of recordings (up to 20\%) on each dataset. For the CBSS algorithm, we will vary the exponent of the loss function (please also see discussion below), and the procedure to prevent convergence to already identified sources (peel-off vs deflationary orthogonalization), as per previous investigations (Avrillon et. al. 2024).
>
> > Additional Feedback  Add some advanced deep learning methods in the benchmark.
>
> Please see reply to ``absence of deep learning baselines" comment above.
>
> > On the Choice of CBSS Objective Function...
>
> The choice of this contrast function is motivated by both theoretical (Klotz and Rohlen, 2025) and empirical findings (Grison et al., 2025), which show that the degree of non-linearity is a key hyperparameter for balancing source separation and overfitting. Higher exponents increase the ability to distinguish between borderline-similar sources but also amplify outliers. Traditional objective functions such as skewness (exponent 3) or kurtosis (exponent 4) offer only a discrete number of available non-linearities. In contrast, the implemented contrast function allows continuous and fine-grained tuning of this property. Notably, traditional functions like skewness can still be recovered by selecting the appropriate exponent (e.g., exponent 3). Indeed, during our experiments, we set the exponent to 3, corresponding to maximizing skewness.
>
> ---
>
> References:
>
> Clarke, A. K., et. al. (2020). Deep learning for robust decomposition of high-density surface EMG signals. IEEE Transactions on Biomedical Engineering, 68(2), 526-534.
> Avrillon, S., et. al. (2024). Tutorial on MUedit: An open-source software for identifying and analysing the discharge timing of motor units from electromyographic signals. Journal of Electromyography and Kinesiology, 77, 102886.
> Grison, A., et. al. (2025). Unlocking the full potential of high‐density surface EMG: novel non‐invasive high‐yield motor unit decomposition. The Journal of Physiology, 603(8), 2281-2300.
> Klotz, T., \& Rohlén, R. (2025). Revisiting convolutive blind source separation for identifying spiking motor neuron activity: From theory to practice. arXiv preprint arXiv:2502.04065.

---

### Decision · Program_Chairs · 2025-09-18

**Decision:**

Accept (poster)

**Comment:**

This submission introduces MUniverse, a modular suite that unifies an EMG simulator interface, curated synthetic/hybrid/experimental datasets with standardized metadata (BIDS/Croissant), internal/external decomposition pipelines, and a common set of benchmarking tasks and metrics to fill the long‑standing gap of open benchmarks for motor‑unit decomposition; reviewers concur that this is a well‑organized resource that addresses a critical need and strengthens reproducibility and comparability in the area. Building on the paper’s strengths noted in review—clear packaging, diverse data sources, and an extensible evaluation framework—concerns centered on missing results for the stated “upper‑bound” estimator, absence of deep‑learning baselines, reliance on expert‑fixed hyperparameters, and some clarity about what MUniverse adds beyond NeuroMotion. Reviewers also flagged presentation issues (initially non‑working notebooks and incomplete Dataverse metadata) and limited subject diversity in one dataset.  In discussion, the authors clarified the higher‑level interface and distinction from NeuroMotion, committed to fix metadata and tutorials and to bring simulation results into the main text, and explicitly stated the resource is not intended for clinical/safety‑critical use; one reviewer considered their critiques largely addressed and was positive about the roadmap, while another acknowledged the value but criticized the rebuttal quality and presentation polish.  Overall, the standardized, open, and practice‑oriented contribution outweighs the limitations, and I recommend Accept (Poster), with camera‑ready requests to report the upper‑bound baseline, include or justify deep‑learning baselines, document hyperparameter choices, improve clarity around the simulator interface, and finalize metadata/tutorials and intended‑use statements.